# AGZO: Activation-Guided Zeroth-Order Optimization for LLM Fine-Tuning

Wei Lin [1]  Yining Jiang [2]  Qingyu Song [2]  Qiao Xiang [2]  Hong Xu [1]

## Abstract

Zeroth-Order (ZO) optimization has emerged as a promising solution for fine-tuning LLMs under strict memory constraints, as it avoids the prohibitive memory cost of storing activations for backpropagation. However, existing ZO methods typically employ isotropic perturbations, neglecting the rich structural information available during the forward pass. In this paper, we identify a crucial link between gradient formation and activation structure: the gradient of a linear layer is confined to the subspace spanned by its input activations. Leveraging this insight, we propose Activation-Guided Zeroth-Order optimization (AGZO). Unlike prior methods, AGZO extracts a compact, activation-informed subspace on the fly during the forward pass and restricts perturbations to this low-rank subspace. We provide a theoretical framework showing that AGZO optimizes a subspace-smoothed objective and provably yields update directions with higher cosine similarity to the true gradient than isotropic baselines. Empirically, we evaluate AGZO on Qwen3 and Pangu models across various benchmarks. AGZO consistently outperforms state-of-the-art ZO baselines and significantly narrows the performance gap with first-order fine-tuning, while maintaining almost the same peak memory footprint as other ZO methods.

## 1. Introduction

Large language models (LLMs) are increasingly adapted to downstream tasks via fine-tuning, but in many practical settings—especially outside large-scale compute clusters—fine-tuning is primarily constrained by GPU memory (Hu et al., 2022; Ouyang et al., 2022). A key reason is that

backpropagation requires storing forward activations (and related intermediate tensors), which can dominate the peak memory footprint at large sequence lengths and batch sizes (Chen et al., 2016; Rajbhandari et al., 2020). Zeroth-order (ZO) optimization provides an appealing alternative in such memory-limited regimes. ZO methods bypass backpropagation by updating parameters using only function evaluations, typically employing randomized finite-difference estimators to approximate the gradient (Nesterov & Spokoiny, 2017; Ghadimi & Lan, 2013; Duchi et al., 2015). Specifically, MeZO adapts two-point randomized finite differences to LLM fine-tuning (Malladi et al., 2023). By employing in-place parameter perturbations and regenerating noise from random seeds, it achieves a peak memory footprint comparable to inference alone. This line of work demonstrates that ZO methods make end-to-end fine-tuning feasible under stricter hardware constraints, substantially reducing training memory while maintaining acceptable performance levels (Sun et al., 2022; Chen et al., 2023; Zhang et al., 2024).

Despite this progress, existing ZO fine-tuning baselines all follow a *black-box perturbation* paradigm: perturbations are sampled from data-independent Gaussian distributions defined solely by parameter dimensions. MeZO uses isotropic full-space perturbations (Malladi et al., 2023), and recent variant LOZO explores low-rank perturbations motivated by spectral properties of gradients (Chen et al., 2025b). Across these approaches, the perturbation distribution is typically independent of the internal representations produced by the current forward pass, which overlooks the structural information exposed by the forward evaluation that is intrinsically linked to the true gradient direction.

Motivated by a simple observation that the weight gradient on a mini-batch is determined by the upstream signals flowing into the layer and the activations produced in the forward pass, we demonstrate that the gradient directions of a linear layer are confined to the subspace spanned by the mini-batch activations, rather than being arbitrary in the full parameter space. Moreover, based on the empirical and theoretical insights that adaptation during fine-tuning is effectively low-dimensional and often admits low-rank structure (Aghajanyan et al., 2021; Li et al., 2018; Hu et al., 2022; Hao et al., 2024), we propose a simple principle for ZO fine-tuning: instead of perturbing weights in unconstrained random directions, one should concentrate perturbations

---

[1]Department of Computer Science and Engineering, The Chinese University of Hong Kong, Hong Kong [2]Xiamen University, China. Correspondence to: Qingyu Song <simmonssong96@gmail.com>.

*Proceedings of the 43rd International Conference on Machine Learning*, Seoul, South Korea. PMLR 306, 2026. Copyright 2026 by the author(s).

within an activation-informed low-dimensional subspace revealed by the forward pass.

Guided by this principle, we propose Activation-Guided Zeroth-Order optimization (AGZO). AGZO samples low-rank, activation-guided perturbations constrained to the subspace, and uses them to form the ZO update. For nonlinear trainable layers, AGZO falls back to standard Gaussian perturbations to preserve general applicability across architectures.

Moreover, AGZO utilizes several strategies to reduce the memory consumption. First, we conduct a lightweight power iteration process in subspace construction (Golub & Van Loan, 2013; Miyato et al., 2018). Second, AGZO keeps compact subspace information and releases in-memory cached activation values immediately after subspace extraction, thereby maintaining the memory advantages of ZO fine-tuning. Compared with existing perturbation baselines, AGZO exploits per-iteration activation structure to construct more informative perturbations, improving update quality while retaining the same memory-efficient character.

We theoretically evaluate the efficacy of our proposed activation-guidance methodology. By analyzing the alignment between the estimated and true gradients, given that most gradient energy lies in the leading singular directions of the activation matrix (Gur-Ari et al., 2018; Papyan, 2020), we prove that AGZO achieves a larger expected cosine similarity to the true gradient than the full-space random perturbation baselines. This result formalizes the intuition that activation-informed subspaces focus the optimization on directions carrying meaningful gradient signals, thereby enhancing the effectiveness of each update step.

We evaluate AGZO on Qwen3 (Yang et al., 2025) and Pangu (Chen et al., 2025a) models under practical GPU memory constraints. AGZO consistently outperforms MeZO and LOZO on various downstream benchmarks, narrowing the gap to first-order fine-tuning. We further support our motivation and theory by directly measuring directional fidelity, where AGZO achieves consistently higher cosine similarity to the true gradients than prior ZO baselines. Finally, we compare peak GPU memory usage by sweeping across sequence lengths and batch sizes, and show that AGZO matches the memory profile of other forward-only ZO baselines while remaining far below first-order training.

The primary contributions of this work are as follows:

- We propose AGZO, a zeroth-order fine-tuning method that extracts compact activation subspaces on the fly and uses them to construct low-rank, activation-guided perturbations. We identify and formalize a fundamental structural link between gradients and activations in linear layers and empirically demonstrate that, during LLM fine-tuning, the gradient signal concentrates in

a low-dimensional subspace revealed by the forward-pass activations.

- We theoretically demonstrate that our proposed activation guidance method improves ZO optimization. We show that AGZO can be viewed as optimizing a subspace-smoothed objective, and its update directions are provably more aligned with the true gradient than random perturbation methods under activation spectral concentration.

- We conduct experiments on various models that jointly demonstrate (i) consistently stronger gradient alignment with the true backpropagation direction, (ii) improved end-to-end fine-tuning performance over prior ZO baselines, and (iii) a peak GPU memory footprint that remains essentially unchanged relative to standard forward-only ZO methods across varying batch size and sequence length.

## 2. Background: Zeroth-Order Fine-Tuning

We consider the standard stochastic optimization problem

$$\min_{W \in \mathbb{R}^d} F(W) \triangleq \mathbb{E}_{B \sim \mathcal{D}}\big[f(W; B)\big], \quad (1)$$

where $W$ denotes the parameters of a LLM, $\mathcal{D}$ is a data distribution over minibatches $B$, and $f(W; B)$ is the empirical loss. First-order methods estimate $\nabla F(W)$ via backpropagation, which requires storing forward activations and thus dominates the training memory footprint (Zhang et al., 2024). Zeroth-order methods approximate gradient directions using only function evaluations and avoid backpropagation, making them attractive for memory-limited fine-tuning of large models (Zhang et al., 2024; Malladi et al., 2023). In this section we briefly review two representative ZO baselines for LLM fine-tuning: MeZO and LOZO (Malladi et al., 2023; Chen et al., 2025b).

### 2.1. Memory-Efficient Full-Space Perturbations

MeZO adapts classical Gaussian-smoothing ZO estimators to the LLM fine-tuning setting. Let $u \sim \mathcal{N}(0, I_d)$ be a standard Gaussian perturbation, organized layer-wise as $u = (U_1, \ldots, U_L)$ where each $U_\ell \in \mathbb{R}^{\frac{d}{L}}$ are layer-wise perturbation parameters. Given a small smoothing parameter $\mu > 0$ and a minibatch $B$, MeZO performs two forward passes to evaluate $f(W + \mu u; B)$ and $f(W; B)$ and constructs the finite difference estimator

$$\widehat{g}_\mu^{\text{MeZO}}(W; B) = \frac{f(W + \mu u; B) - f(W; B)}{\mu} u. \quad (2)$$

This estimator can be interpreted as the gradient of a Gaussian-smoothed objective $F_\mu(W) = \mathbb{E}_u[F(W + \mu u)]$ and is used as a surrogate gradient in a standard optimizer (Nesterov & Spokoiny, 2017).

To remain memory efficient, MeZO never stores the full perturbation tensor $u$ explicitly: it perturbs parameters in place and records only the random seed needed to regenerate $u$ when forming Eq (2). This design reduces fine-tuning memory by roughly a factor of four relative to first-order methods while maintaining competitive performance on downstream tasks (Zhang et al., 2024; Gautam et al., 2024). However, $u$ is supported on the entire parameter space, and empirical studies suggest that layer-wise gradients in LLMs are effectively low-rank (Aghajanyan et al., 2021; Li et al., 2018; Chen et al., 2025b). Full-space isotropic perturbations may spend a substantial portion of the query budget exploring directions that carry little gradient energy.

## 2.2. Low-Rank Zeroth-Order Perturbations

LOZO aims to better match the observed low-rank structure of gradients by introducing a low-rank ZO estimator (Chen et al., 2025b). For each layer $\ell$, LOZO samples random Gaussian factors $U_\ell \in \mathbb{R}^{d_{\text{out}} \times r_\ell}$ and $V_\ell \in \mathbb{R}^{d_{\text{in}} \times r_\ell}$ with $r_\ell \ll \min\{d_{\text{out}}, d_{\text{in}}\}$, and forms the rank-$r_\ell$ perturbation

$$\Delta_\ell = U_\ell V_\ell^\top, \quad (3)$$

so that the full perturbation is $\Delta = (\Delta_1, \ldots, \Delta_L)$. LOZO defines the low-rank gradient estimator as

$$\widehat{g}_\mu^{\text{LOZO}}(W; B) = \frac{f(W + \mu\Delta; B) - f(W; B)}{\mu} \frac{\Delta}{r}, \quad (4)$$

where $r = \{r_\ell\}_{\ell=1}^L$ and the division is understood layer-wise as $\Delta_\ell/r_\ell$. Compared with MeZO, LOZO enforces a low-rank structure that more closely resembles FO gradients in LLM fine-tuning.

## 2.3. Limitations of Existing ZO Baselines

In both MeZO and LOZO, the perturbation distribution is determined entirely by parameter shapes and random seeds. The isotropic directions $u$ in MeZO and the low-rank factors $U_\ell, V_\ell$ in LOZO are sampled from fixed distributions and remain independent of what happens inside the network during the forward pass.

This raises a natural question: can we leverage the intermediate information produced by the forward pass to construct more informative perturbation directions, and hence better zeroth-order gradient approximations? In the next section we analyze the relationship between gradients and activations in LLMs and use these insights to derive design principles for our activation-guided ZO method.

## 3. Structural Analysis of Gradients and Activations

This section analyzes the structural properties of gradients in linear layers, revealing both deterministic links to forward activations and inherent low-rank characteristics. We show that (i) gradients of linear layers admit a simple matrix factorization involving the activation matrices, (ii) both gradients and activations exhibit strong low-rank behavior, and (iii) the gradient row-space is almost entirely contained in the activation column space. These observations motivate constructing zeroth-order perturbations inside an *activation-informed* low-rank subspace rather than in the full parameter space.

### 3.1. Gradient Confinement in Activation Subspaces

We focus on linear layers in LLMs, such as the projection matrices within self-attention mechanisms and fully connected layers in feed-forward networks. These layers dominate the model scale, accounting for most of the trainable parameters in many architectures (Kaplan et al., 2020; Vaswani et al., 2017; Yang et al., 2025).

In Transformer-based architectures, the training data consists of minibatches of sequences. Let $b$ denote the batch size and $T$ the sequence length. While the model processes data as sequences, we can flatten the batch and sequence dimensions into a single dimension $m = b \times T$, representing the total number of tokens in the minibatch. For a linear layer $\ell$ with weight matrix $W_\ell \in \mathbb{R}^{d_{\text{out}} \times d_{\text{in}}}$, we define the aggregated input activation matrix $H_\ell \in \mathbb{R}^{d_{\text{in}} \times m}$ by concatenating the activation vectors of all $m$ tokens. Similarly, let $Q_\ell \in \mathbb{R}^{d_{\text{out}} \times m}$ denote the matrix of upstream gradients with respect to the layer's pre-activations.

Standard backpropagation computes the gradient of the loss with respect to $W_\ell$ by aggregating contributions across all tokens, which takes the compact matrix form:

$$\nabla_{W_\ell} f(W; B) = Q_\ell H_\ell^\top. \quad (5)$$

This factorization reveals a fundamental geometric property: the gradient matrix is formed by a linear combination of the columns of $H_\ell$, i.e., the row-space of the layer-wise gradient is strictly contained in the subspace spanned by the input activations:

$$\text{row}\big(\nabla_{W_\ell} f(W; B)\big) \subseteq \text{col}\big(H_\ell\big). \quad (6)$$

We further quantify how tightly the gradient concentrates in this activation subspace. Fig. 1(a) plots a bar chart of the cosine similarity between the true gradient and its orthogonal projection onto the subspace spanned by the forward activations. Results are obtained when fine-tuning GPT-2 (Radford et al., 2019) on the SST-2 dataset (Wang et al., 2018). Concretely, we perform an SVD of the activation matrix $H_\ell$ and use the leading $r$ singular vectors to define a rank-$r$ activation subspace; we then project the gradient onto this subspace and compute the cosine similarity between the original gradient and the projected one. We report results

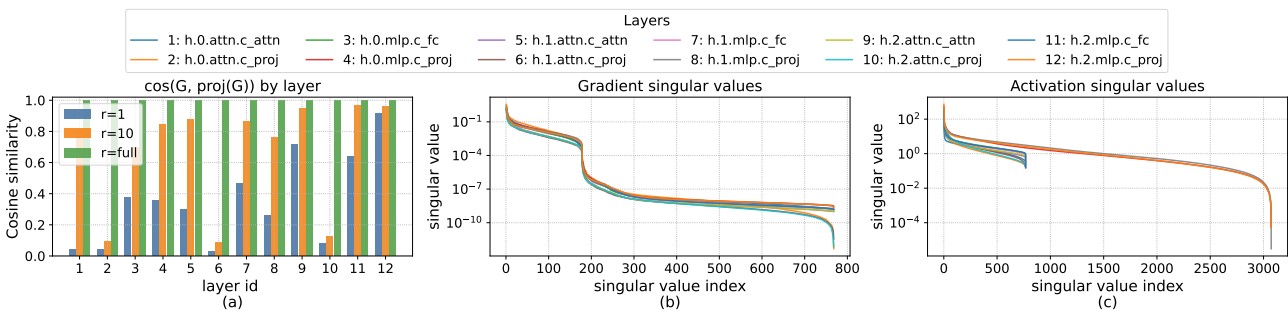

*Figure 1.* Structural analysis of gradients and activations. (a) Cosine similarity between the true gradient and its projection onto the activation subspace. (b) & (c) Singular value spectra of gradients and activations.

for $r = 1, 10$ and for the full activation subspace ($r = 750$). Across layers, the cosine similarity is typically close to 1 when $r \geq 10$, indicating that almost all gradient energy lies in the subspace spanned by the forward activations.

### 3.2. Low-Rank Structure of Gradients and Activations

The matrix factorization in Eq. (5) reveals the structural dependency of gradients on forward activations. While the weight matrices $W_\ell$ reside in high-dimensional spaces, empirical observations suggest that the actual information content typically concentrates in a much smaller subspace.

To examine this structure more concretely, we compute the singular values of $\nabla_{W_\ell} f(W; B)$ via SVD and plot them (on a log scale) for a few representative layers and training steps. Fig. 1(b) shows that the singular values decay rapidly: the spectrum is far from flat, and a small number of leading singular directions dominate the rest. This supports the view that layer-wise gradients are effectively low-rank.

We observe a similar spectral phenomenon for the forward activation matrices. For each $H_\ell$, we compute its singular values and visualize them in Fig. 1(c). Again, the singular values exhibit pronounced decay, indicating that the majority of the activation energy is concentrated along a few dominant directions.

These results show that layer-wise gradient information is concentrated in a low-dimensional subspace that is almost entirely determined by the corresponding activation matrix, and that this activation subspace itself has a rapidly decaying spectrum and can be captured by a small number of leading directions. From a zeroth-order perspective, it is therefore natural to restrict perturbations to a low-rank subspace extracted from forward activations, rather than sampling arbitrary directions in the full parameter space. In the next section we instantiate this idea in an activation-guided ZO method that constructs perturbations inside such activation-informed subspaces.

## 4. Activation-Guided Zeroth-Order Optimization

We now introduce Activation-Guided Zeroth-Order optimization (AGZO). For linear layers, AGZO perturbs weights inside an activation-guided low-rank subspace; for nonlinear layers, AGZO simply uses Gaussian perturbations. Each iteration reuses the standard forward pass to both evaluate the loss and extract dominant activation directions, without storing activation matrices across iterations.

### 4.1. AGZO Algorithm

Consider a linear layer $\ell$ with weight matrix $W_\ell \in \mathbb{R}^{d_{\text{out}} \times d_{\text{in}}}$ and activation matrix $H_\ell \in \mathbb{R}^{d_{\text{in}} \times m}$ for the current mini-batch, as in Section 3.1. AGZO constructs, on the fly, an activation-informed subspace from $H_\ell$ and samples perturbations inside this subspace.

**Activation-informed subspace.** Given a target rank $r \ll \min\{d_{\text{out}}, d_{\text{in}}, m\}$, we approximate the top $r$ left singular vectors of $H_\ell$ via a few steps of power iteration on $H_\ell H_\ell^\top$, using only matrix–matrix products with $H_\ell$ and $H_\ell^\top$ (Bentbib & Kanber, 2015). The routine in Algorithm 1 takes the current activation matrix $H$ and returns an orthonormal basis $A \in \mathbb{R}^{d_{\text{in}} \times r}$ whose columns span a rank-$r$ subspace of $\text{col}(H)$. Once $A_\ell$ is computed for the current minibatch, we discard $H_\ell$ to reduce memory consumption.

**Perturbations and zeroth order estimator.** Given $A_\ell \in \mathbb{R}^{d_{\text{in}} \times r_\ell}$, AGZO samples a low-rank perturbation for each linear layer by drawing a left factor $R_\ell \in \mathbb{R}^{d_{\text{out}} \times r_\ell}$ with i.i.d. standard normal entries and setting

$$\Delta_\ell = \begin{cases} R_\ell A_\ell^\top, & \text{if layer } \ell \text{ is linear,} \\ u_\ell, & \text{if layer } \ell \text{ is nonlinear,} \end{cases} \quad (7)$$

where $u_\ell$ is a Gaussian perturbation with the same shape as $W_\ell$. The full perturbation is then $\Delta = (\Delta_1, \ldots, \Delta_L)$. For linear layers, each $\Delta_\ell$ has rank at most $r$, and its row space is contained in the activation-informed subspace spanned by $A_\ell$.

**Algorithm 1** SUBSPACEEXTRACT($H, r, K$): activation-informed basis via power iteration

---

1: **Input:** activation matrix $H \in \mathbb{R}^{d_{in} \times m}$, target rank $r$, number of power-iteration steps $K$
2: Sample Gaussian test matrix $\Omega \in \mathbb{R}^{m \times r}$
3: $Y \leftarrow H\Omega$
4: **for** $k = 1, \ldots, K$ **do**
5:    $[Q, \sim] \leftarrow \mathrm{qr}(Y)$       // orthonormalize columns
6:    $Y \leftarrow H(H^\top Q)$
7: **end for**
8: $[Q, \sim] \leftarrow \mathrm{qr}(Y)$
9: **return** $A \leftarrow Q$        // $A \in \mathbb{R}^{d_{in} \times r}$

---

Given a smoothing parameter $\mu > 0$ and minibatch $B$, we first evaluate

$$f_0 = f(W; B), \tag{8}$$

and, during this forward pass, compute $\{A_\ell\}$ for all linear layers using Algorithm 1. We then form $\Delta$ via (7), evaluate the perturbed loss

$$f_+ = f(W + \mu\Delta; B), \tag{9}$$

and define the layer-wise estimator

$$\widehat{\nabla}_{W_\ell} f^{\mathrm{AGZO}}(W; B) = \frac{f_+ - f_0}{\mu} \Delta_\ell, \quad \ell = 1, \ldots, L. \tag{10}$$

Stacking these matrices across layers yields $\widehat{g}_\mu^{\mathrm{AGZO}}(W; B)$, which is used in a ZO gradient descent update.

Algorithm 2 summarizes one AGZO iteration. Subspace extraction for each layer is done immediately when its activation matrix becomes available in the forward pass, so full activations are never stored beyond this step.

In practice, a small number of power-iteration steps ($K = 3$) per layer is enough to get satisfactory approximation, which adds only a few matrix multiplications on top of the forward pass. The dominant cost per iteration is thus forward evaluations, as in MeZO and LOZO.

### 4.2. Memory Usage Analysis

We analyze the memory footprint of each method, focusing on the *optimization overhead*—defined as the storage required beyond the fixed model parameters. Both MeZO and LOZO incur essentially no overhead, as their perturbations (whether isotropic or low-rank factors) are generated from random seeds and can be regenerated on the fly. AGZO requires storing the activation-informed basis $A_\ell \in \mathbb{R}^{d_{in} \times r}$ for each layer, as it depends on the input data and cannot be recovered from a seed. However, this overhead is negligible compared to the model size: for a weight matrix $W_\ell$ with $d_{out} \times d_{in}$ parameters, the basis $A_\ell$ requires only $d_{in} \times r$.

**Algorithm 2** AGZO Iteration

---

1: **Input:** Weights $W$, ranks $\{r_\ell\}$, scalars $\mu, \eta, K$.
2: **1. Forward & Subspace Extraction (via Hooks):**
3: Run forward pass on batch $B$ to compute $f_0$.
4: **During** computation at each linear layer $\ell$: Extract $A_\ell \leftarrow$ SUBSPACEEXTRACT($H_\ell, r_\ell, K$).
5: **2. In-Place Perturbation:**
6: **for** layer $\ell = 1 \ldots L$ **do**
7:    Sample random seed $s_\ell$.
8:    **if** layer $\ell$ is linear **then**
9:       Generate $R_\ell \sim \mathcal{N}(0, I)$ from $s_\ell$;   Set update matrix $\Delta_\ell = R_\ell A_\ell^\top$.
10:    **else**
11:       Generate $\Delta_\ell \sim \mathcal{N}(0, I)$ from $s_\ell$.
12:    **end if**
13:    $W_\ell \leftarrow W_\ell + \mu\Delta_\ell$    {Apply perturbation in-place}
14: **end for**
15: **3. Gradient Estimate & Update:**
16: Compute $f_+ = f(W; B)$ with perturbed weights.
17: Set projected gradient scalar $g \leftarrow (f_+ - f_0)/\mu$.
18: **for** layer $\ell = 1 \ldots L$ **do**
19:    Regenerate $\Delta_\ell$ using stored seed $s_\ell$ (and $A_\ell$ if linear).
20:    $W_\ell \leftarrow W_\ell - \mu\Delta_\ell - \eta \cdot g \cdot \Delta_\ell$    {Restore & Update}
21: **end for**

---

With $r \ll d_{out}$, this consumes a tiny fraction of the memory needed for the weights themselves.

In our experiments, we set $r = 1$. This choice minimizes the storage overhead for the basis $A_\ell$ to the theoretical lower bound. More importantly, since the AGZO perturbation is stochastic within the subspace spanned by $A_\ell$, restricting the rank to 1 forces the random exploration to concentrate entirely on the single most dominant direction of the activation energy. This prevents the update signal from being diluted across less significant components.

## 5. Provable Superiority of AGZO

We now analyze the AGZO estimator introduced in section 4. The goal is to understand (i) its mathematical essence and how far it is from the true gradient, and (ii) how its directional quality compares to MeZO. We focus on linear layers where AGZO uses low-rank perturbations as in Eq. (7) and the zeroth order estimator in Eq. (10).

### 5.1. AGZO is A Projected Gradient Estimator

We first show that AGZO can be interpreted as estimating a projected gradient of a subspace-smoothed objective.

Condition on the subspace bases $A := \{A_\ell\}$ computed at the current iterate $W$. Let $\Delta(W, R)$ be the random perturbation defined in Eq. (7) and define the subspace-smoothed

objective

$$F_{\mu,A}(W) := \mathbb{E}_R\big[F\big(W + \mu\,\Delta(W, R)\big)\big]. \quad (11)$$

The next proposition shows that, up to smoothing, AGZO is an exact gradient estimator projected onto the activation-informed subspace.

**Proposition 5.1.** *Assume $F$ has $L$-Lipschitz gradient. The AGZO estimator satisfies, for each linear layer $\ell$,*

$$\mathbb{E}_{R,B}\Big[\widehat{\nabla}_{W_\ell}^{\mathrm{AGZO}}(W; B)\Big] = \nabla_{W_\ell} F_{\mu,A}(W)\,A_\ell A_\ell^\top. \quad (12)$$

*In particular, AGZO estimates the gradient of the subspace-smoothed objective (11), projected onto $\mathrm{span}(A_\ell)$.*

*Proof.* See Theorem A.3(b) in Appendix. □

Under standard smoothness assumptions, the difference between $\nabla F_{\mu,A}$ and $\nabla F$ after projection vanishes linearly as $\mu \to 0$.

**Proposition 5.2.** *Suppose $F$ has $L$-Lipschitz gradient. Then for each layer $\ell$ there exists a constant $C_\ell > 0$, depending only on $L$ and the layer dimensions, such that*

$$\big\|\nabla_{W_\ell} F_{\mu,A}(W)\,A_\ell A_\ell^\top - \nabla_{W_\ell} F(W)\,A_\ell A_\ell^\top\big\|_F \ \le\ C_\ell\,\mu. \quad (13)$$

*Proof.* See Theorem A.3(c) in Appendix. □

The remaining component of the bias comes from projecting $\nabla_{W_\ell} F(W)$ onto $\mathrm{span}(A_\ell)$. Recall the gradient factorization (5) and the subspace analysis (6). In AGZO, $A_\ell$ is constructed to approximate a low-rank activation subspace for $\mathrm{col}(H_\ell)$. For exposition, consider the idealized case where this subspace exactly supports the gradient.

**Corollary 5.3.** *Suppose for a given layer $\ell$ and all mini-batches $B$,*

$$\mathrm{row}\big(\nabla_{W_\ell} f(W, B)\big) \subseteq \mathrm{span}(A_\ell), \quad (14)$$

*so that $\nabla_{W_\ell} F(W) = \nabla_{W_\ell} F(W)\,A_\ell A_\ell^\top$. Then combining (12) and (13) yields*

$$\Big\|\mathbb{E}_{R,B}\big[\widehat{\nabla}_{W_\ell}^{\mathrm{AGZO}}(W; B)\,\big|\,A\big] - \nabla_{W_\ell} F(W)\Big\|_F \ \le\ C_\ell\,\mu. \quad (15)$$

*Thus, in this regime AGZO is an asymptotically unbiased estimator of the true layer gradient as $\mu \to 0$.*

In practice, the activation-guided subspace only approximates the row space. Section 3 shows that the overlap between $\nabla_{W_\ell} F(W)$ and $\nabla_{W_\ell} F(W)\,A_\ell A_\ell^\top$ is nevertheless very close to one if the approximation rank is high enough.

## 5.2. Directional Fidelity of AGZO

Since the update length can be tuned by step size, the effectiveness of a gradient estimator mainly depends on its directional quality: how well its direction aligns with the true gradient. This subsection analyzes the expected cosine similarity between the estimator and the true gradient, and compares AGZO with MeZO in a noiseless setting.

Let $G \in \mathbb{R}^{d_{\mathrm{out}} \times d_{\mathrm{in}}}$ denote the true gradient $\nabla_{W_\ell} F(W, B)$, and let $\widehat{G}$ be the approximated gradient. Define the cosine similarity:

$$\cos(\widehat{G}, G) := \frac{\langle \widehat{G}, G \rangle_F}{\|\widehat{G}\|_F\,\|G\|_F}, \quad \langle \widehat{G}, G \rangle_F := \mathrm{tr}(\widehat{G}^\top G). \quad (16)$$

We analyze the expected cosine similarity in a noiseless setting ($\mu \to 0$) where the finite difference oracle returns exact directional derivatives and stochastic minibatch noise is ignored.

**Theorem 5.4.** *Let $G \in \mathbb{R}^{d_{\mathrm{out}} \times d_{\mathrm{in}}}$ be fixed and $A \in \mathbb{R}^{d_{\mathrm{in}} \times r}$ have orthonormal columns. Let $\widehat{G}_0^{\mathrm{AGZO}}$ be the noiseless AGZO estimator, which has the form:*

$$\widehat{G}_0^{\mathrm{AGZO}} = \langle G, \Delta \rangle_F\,\Delta = \langle G, RA^\top \rangle_F\,RA^\top. \quad (17)$$

*Then*

$$\mathbb{E}_R\big[\cos\big(\widehat{G}_0^{\mathrm{AGZO}}, G\big)\big] = \beta_{d_{\mathrm{out}}r}\,\frac{\|GA\|_F}{\|G\|_F}, \quad (18)$$

*where*

$$\beta_D = \frac{\Gamma\big(\frac{D}{2}\big)}{\sqrt{\pi}\,\Gamma\big(\frac{D+1}{2}\big)}, \quad (19)$$

*and for any $D \ge 2$, $\beta_D$ satisfies the tight bounds:*

$$\sqrt{\frac{2}{\pi D}} \ \le\ \beta_D \ \le\ \sqrt{\frac{2}{\pi(D-1)}}. \quad (20)$$

*Proof.* See Appendix A.2.1. □

The factor $\|GA\|_F/\|G\|_F$ has a natural geometric interpretation: it is precisely the fraction of gradient Frobenius energy captured by the AGZO subspace, since $\|GA\|_F = \|GAA^\top\|_F$ (See remark A.6 in Appendix). Thus AGZO benefits both from working in a lower effective dimension $mr$ (through $\beta_{mr}$) and from aligning its perturbation subspace with directions where $G$ has large energy.

For the MeZO baseline with Gaussian perturbations, the estimator has the same form but with $A = I_{d_{\mathrm{in}}}$ and $\Delta = R \in \mathbb{R}^{d_{\mathrm{out}} \times d_{\mathrm{in}}}$ dense. Theorem 5.4 then yields the following corollary.

**Corollary 5.5.** *Consider the noiseless MeZO estimator $\widehat{G}_0^{\mathrm{MEZO}}$ constructed from dense Gaussian directions $\Delta = R$ with $R \sim \mathcal{N}(0, I_{d_{out} \times d_{in}})$. Then*

$$\mathbb{E}_R\big[\cos\big(\widehat{G}_0^{\mathrm{MEZO}}, G\big)\big] = \beta_{d_{out}d_{in}}. \tag{21}$$

*This corresponds to the special case of Theorem 5.4 with $A = I_{d_{in}}$ and $\|GA\|_F/\|G\|_F = 1$.*

To compare AGZO and MeZO explicitly, we analyze their expected cosine similarity to the true gradient. Consider a layer with gradient factorization $\nabla_{W_\ell} F(W) = Q_\ell H_\ell^\top$. Let the compact SVD of the activation matrix be $H_\ell = U_\ell \Sigma_\ell V_\ell^\top$. We define the interaction matrix between the upstream gradient and activation subspaces as:

$$B_\ell := V_\ell^\top Q_\ell^\top Q_\ell V_\ell \succeq 0.$$

This matrix captures the energy distribution of the gradient. Specifically, the diagonal entry $B_{\ell,ii}$ quantifies the energy of the upstream gradient projected onto the $i$-th principal component of the activation inputs.

We now state our main result. It demonstrates that unless the gradient signal is adversarially aligned with the smallest singular values of the activation, AGZO provably outperforms MeZO.

**Theorem 5.6.** *Consider a layer where the activation matrix $H_\ell$ is low-rank (i.e., rank $s_\ell < d_{in}$). Assume the upstream gradient energy is broadly distributed such that the average energy along the top-$r_\ell$ directions is not less than the global average:*

$$\frac{1}{r_\ell} \sum_{i=1}^{r_\ell} B_{\ell,ii} \ge \frac{1}{s_\ell} \sum_{i=1}^{s_\ell} B_{\ell,ii}. \tag{22}$$

*Then, the AGZO provably yields a higher expected cosine similarity to the true gradient than MeZO :*

$$\mathbb{E}_R\big[\cos\big(\widehat{G}_0^{\mathrm{AGZO}}, G_\ell\big)\big] > \mathbb{E}_R\big[\cos\big(\widehat{G}_0^{\mathrm{MEZO}}, G_\ell\big)\big]. \tag{23}$$

*Furthermore, the performance gap widens as the activation singular values become more heterogeneous (i.e., faster decay).*

*Proof.* See Appendix A.3. □

Theorem 5.6 formalizes the intuition behind AGZO: for layers where gradients concentrate in low-rank activation subspaces, AGZO produces update directions that are significantly better aligned with the true gradient than dense isotropic baselines.

## 6. Experiments

This section evaluates AGZO on fine-tuning LLMs under practical memory constraints. The experiments cover multiple tasks, including the SuperGLUE benchmark (Wang

*Table 1.* Experiments on Qwen3-0.6b. Bold: ZO's best results.

| Task | FO | AGZO | MeZO | LOZO | Zero | ICL |
|------|------|------|------|------|------|------|
| SST2 | 0.904 | **0.877** | 0.858 | 0.870 | 0.540 | 0.510 |
| COPA | 0.730 | **0.740** | 0.680 | 0.690 | 0.570 | 0.620 |
| CB | 0.946 | **0.892** | 0.803 | 0.760 | 0.410 | 0.570 |
| BoolQ | 0.768 | 0.724 | **0.730** | 0.724 | 0.646 | 0.700 |
| MultiRC | 0.826 | **0.756** | 0.734 | 0.737 | 0.518 | 0.673 |
| RTE | 0.808 | **0.772** | 0.732 | 0.743 | 0.599 | 0.722 |
| WiC | 0.675 | **0.595** | 0.573 | 0.575 | 0.498 | 0.523 |
| SQuAD | 0.871 | **0.790** | 0.779 | 0.785 | 0.416 | 0.414 |

*Table 2.* Experiments on Qwen3-4b.

| Task | AGZO | MeZO | LOZO | Zero | ICL |
|------|------|------|------|------|------|
| SST2 | **0.892** | 0.875 | 0.866 | 0.649 | 0.887 |
| CB | **0.875** | 0.857 | 0.857 | 0.375 | 0.821 |
| BoolQ | 0.820 | 0.823 | 0.822 | 0.790 | **0.827** |
| MultiRC | **0.853** | 0.850 | 0.852 | 0.765 | 0.849 |
| RTE | **0.848** | 0.837 | 0.801 | 0.805 | 0.835 |
| WiC | **0.678** | 0.666 | 0.659 | 0.595 | 0.615 |
| SQuAD | **0.876** | 0.870 | 0.869 | 0.583 | 0.555 |

et al., 2019) and other datasets. We conduct our evaluation on the Qwen3 family (0.6B and 4B scales) (Yang et al., 2025) and the openPangu (Chen et al., 2025a) model. In particular, we select OPENPANGU-EMBEDDED-1B (Rang et al., 2025) (denoted as Pangu-1B) [1], a model specifically designed for efficient inference on edge devices (we use the GPU variant in our experiments[2]).

We compare AGZO against established zeroth-order baselines (MeZO, LOZO) as well as non-training baselines (zero-shot prompting and in-context learning, denoted as ICL). Additionally, we include a first-order fine-tuning baseline (FO) using standard backpropagation whenever memory constraints allow. ZO optimizers are fine-tuned for 20,000 steps, whereas the FO baseline is trained for 1,000 steps.

We build two testbeds with different computation platforms.

1. **Testbed one** is an Ubuntu machine equipped with two NVIDIA RTX 3090 GPUs.

2. **Testbed two** is an EulerOS machine equipped with eight Ascent 910B2 NPUs.

To ensure a fair comparison, all ZO methods update the full set of trainable parameters and share identical data preprocessing and evaluation pipelines. Regarding hyperparameters, we fix the smoothing parameter $\mu$ to $1 \times 10^{-7}$ for all ZO methods on Qwen3 and set $\mu = 1 \times 10^{-4}$ for Pangu-1B, while the learning rate is determined via grid search based on validation set performance. Further experimental details are provided in Appendix B.

---

[1]https://huggingface.co/FreedomIntelligence/openPangu-Embedded-1B

[2]https://modelscope.cn/models/wangrongsheng/openPangu-Embedded-1B-model-GPU

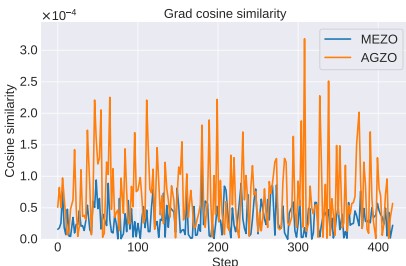
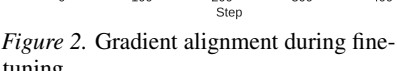

*Figure 2.* Gradient alignment during fine-tuning.

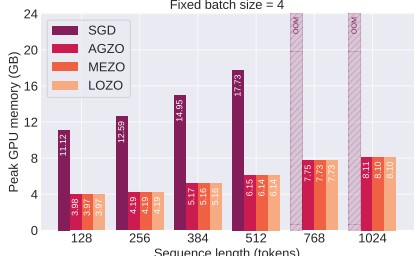
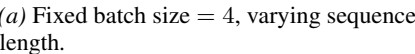

*(a)* Fixed batch size $= 4$, varying sequence length.

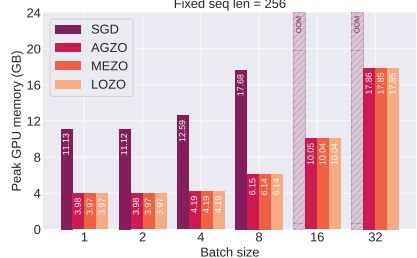

*(b)* Fixed sequence length $= 256$, varying batch size.

*Figure 3.* Peak GPU memory usage when fine-tuning Qwen3-0.6B on DROP.

*Table 3.* Experiments on Pangu-1B.

| Task | FO | AGZO | MeZO | LOZO | Zero | ICL |
|------|------|-------|-------|-------|-------|-------|
| SST2 | 0.822 | **0.778** | 0.764 | 0.720 | 0.568 | 0.717 |
| COPA | 0.800 | **0.770** | 0.750 | **0.770** | 0.760 | 0.750 |
| CB | 0.696 | **0.732** | **0.732** | 0.679 | 0.500 | 0.446 |
| BoolQ | 0.751 | **0.730** | 0.699 | 0.696 | 0.695 | 0.735 |
| RTE | 0.780 | **0.736** | 0.729 | 0.697 | 0.581 | 0.682 |
| WiC | 0.657 | **0.575** | 0.567 | 0.563 | 0.466 | 0.511 |

## 6.1. Alignment to the True Gradient

This diagnostic experiment is designed to validate the gradient accuracy analysis in Section 5, which shows that under activation spectral concentration, AGZO achieves a strictly larger expected cosine similarity than MeZO.

To empirically test this prediction in a realistic fine-tuning setting, we fine-tune QWEN3-0.6B with Testbed One on SST-2 and track, at training step $t$, the cosine similarity between the ZO estimated gradient and the exact backpropagation gradient computed on the same mini-batch. As shown in Figure 2, while the absolute cosine values are small due to the extremely large parameter dimension and the stochasticity of fine-tuning, the persistent gap between AGZO and MeZO is the primary signal of interest and is consistent with the strict separation predicted by Theorem 5.6.

## 6.2. End-to-End Fine-Tuning Performance (Testbed One)

**QWEN3-0.6B.** Table 1 shows that AGZO achieves consistently stronger downstream performance than existing ZO baselines across a broad set of benchmarks. By producing update directions that are better aligned with the true gradient, AGZO enables more effective optimization under the same query budget. As a result, AGZO converges to better solutions and noticeably narrows the performance gap between zeroth-order fine-tuning and first-order training.

**QWEN3-4B.** Table 2 reports the results on Qwen3-4B. Under the same hardware setting, FO method runs out of memory, whereas ZO methods remain feasible. AGZO consistently outperforms MeZO and LOZO on this larger scale. AGZO narrow down the gap between memory efficiency

and optimization quality, enabling effective fine-tuning of larger models on consumer-grade GPUs.

**PANGU-1B.** Table 3 summarizes the performance of AGZO and various baselines on the Pangu-1B model. Overall, AGZO consistently outperforms existing zeroth-order baselines (MEZO and LOZO) and non-training baselines (zero-shot prompting and in-context learning) on most tasks, demonstrating the effectiveness of our approach in adapting large models with limited gradient information.

## 6.3. End-to-End Cross-Platform Fine-Tuning Performance

We evaluate the cross-platform inference performance, i.e, evaluating the performance on NPU (with Testbed Two) with GPU-trained models (with Testbed One). Table 4 presents the performance of AGZO and various baselines on OPENPANGU-EMBEDDED-1B across on NPU. Across both GPU and NPU, AGZO consistently achieves the best results among zeroth-order methods and non-training baselines on most downstream tasks. On the NPU, AGZO attains an average score of 0.709, outperforming other ZO baselines on tasks such as SST2, COPA, BoolQ, RTE, and WiC. The slightly lower performance on the NPU compared to the GPU may be attributed to subtle differences in numerical precision, memory layout, or low-level kernel implementations, which can affect the propagation of small perturbations used in zeroth-order optimization.

*Table 4.* Experiments on Pangu-1B(NPU). The best results are shown in bold except for FO.

| Task | FO | AGZO | MeZO | LOZO | Zero | ICL |
|------|------|-------|-------|-------|-------|-------|
| SST2 | 0.821 | 0.765 | **0.766** | 0.718 | 0.571 | 0.710 |
| COPA | 0.800 | **0.770** | 0.740 | 0.720 | 0.760 | 0.740 |
| CB | 0.696 | 0.696 | **0.732** | 0.643 | 0.482 | 0.446 |
| BoolQ | 0.752 | **0.728** | 0.697 | 0.694 | 0.696 | 0.731 |
| RTE | 0.780 | **0.729** | **0.729** | 0.682 | 0.578 | 0.682 |
| WiC | 0.657 | **0.567** | 0.552 | 0.542 | 0.469 | 0.495 |
| Avg. | 0.738 | **0.709** | 0.703 | 0.667 | 0.593 | 0.636 |

## 6.4. Peak GPU Memory Footprint

As discussed in Section 4.2, AGZO only stores the activation-informed basis for each linear layer, without maintaining additional activation state beyond standard ZO state. Since this basis is tiny compared to the weight matrix, AGZO incurs nearly the same peak GPU memory as MEZO.

We empirically validate this memory analysis by measuring the *peak* GPU memory footprint when fine-tuning Qwen3-0.6B on DROP task with Testbed one. We sweep the two primary drivers of training-time memory: sequence length and batch size. In Figure 3(a), we fix the batch size to 4 and increase the sequence length. FO exhibits rapidly growing memory usage and becomes out-of-memory (OOM) at long contexts, whereas ZO methods remain substantially lower and continue to run. In Figure 3(b), we fix the sequence length to 256 and increase the batch size. FO again hits OOM at moderate batch sizes, while ZO methods remain feasible for significantly larger batches.

Importantly, AGZO matches the memory profile of other forward-only ZO baselines, indicating that the activation-guided subspace construction introduces negligible additional memory overhead.

## 7. Conclusions

In this paper, we propose AGZO, a zeroth-order fine-tuning method that leverages per-iteration activation structure to construct low-rank, activation-guided perturbations for linear layers. By extracting compact activation subspaces on the fly via lightweight power iteration, AGZO concentrates ZO updates on directions that are intrinsically coupled to backpropagation signals, all without storing activations across iterations. Theoretically, we prove that under activation spectral concentration, the AGZO update direction achieves a strictly larger expected cosine similarity to the true gradient than prior isotropic ZO baselines. Experiments on Qwen3 and Pangu models support this analysis, showing that AGZO generally improves over the considered ZO baselines in downstream performance and relative directional fidelity, while maintaining a peak memory footprint comparable to standard forward-only ZO methods.

## Impact Statement

This paper presents work whose goal is to advance the field of Machine Learning. There are many potential societal consequences of our work, none which we feel must be specifically highlighted here.

## Acknowledgements

This work is supported in part by funding from CUHK (4937007, 4937008, 5501329, 5501517, 8601129) and the Key Project for Technological Innovation and Industrialization in the Software Industry of Fujian Province (Project "Research, Development, and Industrialization of Key Technologies for AI Agent-Driven Production Management in the Electronic Manufacturing Industry").

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

# Appendix

## A. Proofs

### A.1. Subspace/Gaussian Smoothing Identities

This section places AGZO and MEZO under one oracle view: each estimator computes (up to $O(\mu^2)$) the gradient of a *smoothed* objective. For AGZO, the smoothing kernel is restricted to a row-subspace; for MEZO it is isotropic.

#### A.1.1. SMOOTHING OPERATORS

Fix a step radius $\mu > 0$. For a given layer $l$, let $A_l \in \mathbb{R}^{d_{in} \times r}$ have orthonormal columns ($A_l^\top A_l = I_r$), and let $R_l \in \mathbb{R}^{d_{out} \times r}$ have i.i.d. $\mathcal{N}(0,1)$ entries. Define the rank-$r$ matrix direction $\Delta_l = R_l A_l^\top$ and the block direction $\Delta = \{\Delta_l\}_{l=1}^L$.

**Per-batch and population smoothings.** For a fixed batch $B$, define the *subspace smoothing* of $f(\cdot, B)$:

$$f_{\mu,A}(W,B) \ := \ \mathbb{E}_R\Big[f\big(W + \mu\Delta, B\big)\Big], \qquad \Delta_l = R_l A_l^\top, \ \ R := \{R_l\}_{l=1}^L. \tag{24}$$

Averaging over batches yields the population version

$$F_{\mu,A}(W) \ := \ \mathbb{E}_R\Big[F\big(W + \mu\Delta\big)\Big] \ = \ \mathbb{E}_{B,R}\Big[f\big(W + \mu\Delta, B\big)\Big]. \tag{25}$$

For MEZO, let $U_l \in \mathbb{R}^{d_{out} \times d_{in}}$ have i.i.d. $\mathcal{N}(0,1)$ entries and set $\Delta_l = U_l$ (full-dimensional). Define

$$f_{\mu,\text{iso}}(W,B) \ := \ \mathbb{E}_U\Big[f\big(W + \mu\Delta, B\big)\Big], \qquad F_{\mu,\text{iso}}(W) \ := \ \mathbb{E}_{B,U}\Big[f\big(W + \mu\Delta, B\big)\Big]. \tag{26}$$

#### A.1.2. SUBSPACE-SMOOTHING GRADIENT IDENTITY FOR AGZO

**Lemma A.1** (Gaussian moment identity). *Let $R \in \mathbb{R}^{d_{out} \times r}$ have i.i.d. $\mathcal{N}(0,1)$ entries and let $M \in \mathbb{R}^{d_{out} \times r}$ be deterministic. Then*

$$\mathbb{E}\left[\langle M, R\rangle\, R\right] \ = \ M, \qquad \text{where } \langle M, R\rangle := \text{tr}(M^\top R).$$

*Similarly, if $U \in \mathbb{R}^{d_{out} \times d_{in}}$ is i.i.d. standard Gaussian and $G \in \mathbb{R}^{d_{out} \times d_{in}}$, then $\mathbb{E}\left[\langle G, U\rangle\, U\right] = G$.*

*Proof.* We proceed entrywise. For $a \in \{1, \ldots, d_{out}\}$ and $b \in \{1, \ldots, r\}$,

$$\big[\mathbb{E}\left[\langle M, R\rangle\, R\right]\big]_{ab} = \mathbb{E}\Big[\Big(\sum_{i=1}^{d_{out}}\sum_{j=1}^{r} M_{ij} R_{ij}\Big) R_{ab}\Big] = \sum_{i=1}^{d_{out}}\sum_{j=1}^{r} M_{ij}\, \mathbb{E}\left[R_{ij} R_{ab}\right].$$

Because $R$ has i.i.d. $\mathcal{N}(0,1)$ entries: (i) $\mathbb{E}[R_{ij}] = 0$ and $\mathbb{E}[R_{ij}^2] = \text{Var}(R_{ij}) = 1$; (ii) if $(i,j) \neq (a,b)$ then $R_{ij}$ and $R_{ab}$ are independent, hence $\mathbb{E}[R_{ij} R_{ab}] = \mathbb{E}[R_{ij}]\,\mathbb{E}[R_{ab}] = 0$. Combining, $\mathbb{E}[R_{ij} R_{ab}] = 1$ when $(i,j) = (a,b)$ and 0 otherwise, we have:

$$\mathbb{E}[R_{ij} R_{ab}] \ = \ \delta_{ia}\, \delta_{jb} \ = \ \mathbf{1}\{i = a,\ j = b\}.$$

Using the claim, the double sum collapses to the single surviving term $M_{ab}$:

$$\big[\mathbb{E}\left[\langle M, R\rangle\, R\right]\big]_{ab} = M_{ab}.$$

Since this holds for every $(a,b)$, we have $\mathbb{E}\left[\langle M, R\rangle\, R\right] = M$.

The isotropic version is identical: for $U \in \mathbb{R}^{d_{out} \times d_{in}}$ i.i.d. $\mathcal{N}(0,1)$ and deterministic $G$,

$$\big[\mathbb{E}\left[\langle G, U\rangle\, U\right]\big]_{ab} = \sum_{i=1}^{d_{out}}\sum_{j=1}^{d_{in}} G_{ij}\, \mathbb{E}[U_{ij} U_{ab}] = G_{ab},$$

so $\mathbb{E}\left[\langle G, U \rangle U\right] = G$.

*Vectorized view.* Let $z = \text{vec}(R) \sim \mathcal{N}(0, I_{d_{out}r})$ and $x = \text{vec}(M)$. Then $\mathbb{E}[(x^\top z)z] = \mathbb{E}[zz^\top]x = Ix = x$, which is the same identity reshaped to matrices. □

**Lemma A.2** (Stein identity for matrix Gaussians). *Let $R \in \mathbb{R}^{d_{out} \times r}$ have i.i.d. $\mathcal{N}(0, 1)$ entries with joint density $p(R) = \prod_{a=1}^{d_{out}} \prod_{b=1}^{r} \phi(R_{ab})$, where $\phi(z) = (2\pi)^{-1/2} e^{-z^2/2}$. Let $h : \mathbb{R}^{d_{out}r} \to \mathbb{R}$ be $C^1$ with $\mathbb{E}|h(R)| < \infty$ and $\mathbb{E}\|\nabla_R h(R)\|_F < \infty$. Assume moreover the following* sub-Gaussian growth *condition:*

*(A) For each coordinate $(a, b)$, writing $U := \{R_{ij} : (i, j) \neq (a, b)\}$ and $g(z; U) := h(R^{(-ab)}, z)$, there exist $\alpha \in (0, \frac{1}{2})$ and a nonnegative random variable $C(U)$ with $\mathbb{E}C(U) < \infty$ such that, for all $z \in \mathbb{R}$,*

$$\left|g(z; U)\right| + \left|\partial g(z; U)/\partial z\right| \leq C(U) e^{\alpha z^2}.$$

*Then*

$$\mathbb{E}\left[h(R) R\right] = \mathbb{E}\left[\nabla_R h(R)\right], \tag{27}$$

*where the expectation is taken entrywise and $\nabla_R h$ is the matrix of partial derivatives of $h$ with respect to the entries of $R$.*

*Proof.* Fix $(a, b)$ and let $Z := R_{ab} \sim \mathcal{N}(0, 1)$, independent of $U := \{R_{ij} : (i, j) \neq (a, b)\}$. Define $g(z; U) := h(R^{(-ab)}, z)$ with $R_l^{(-ab)}$ the matrix of fixed other entries, we have $h(R) = g(Z; U)$.

Conditioning on $U$ and using independence of $Z$ and $U$,

$$\mathbb{E}\left[h(R) R_{ab}\right] = \mathbb{E}_U\left[\mathbb{E}_Z\left[g(Z; U) Z\right]\right] = \mathbb{E}_U\left[\int_{\mathbb{R}} g(z; U) z \, \phi(z) \, dz\right].$$

For fixed $U$ and $M > 0$,

$$\int_{-M}^{M} g(z; U) z \, \phi(z) \, dz = -\int_{-M}^{M} g(z; U) \, \phi'(z) \, dz = -\left[g(z; U)\phi(z)\right]_{-M}^{M} + \int_{-M}^{M} g'(z; U) \, \phi(z) \, dz,$$

since $\phi'(z) = -z\phi(z)$. By (A), $|g(z; U)\phi(z)| \leq C(U)(2\pi)^{-1/2}e^{-(\frac{1}{2}-\alpha)z^2} \to 0$ as $|z| \to \infty$, so the boundary term vanishes as $M \to \infty$. Dominated convergence (dominated by $C(U)e^{-(\frac{1}{2}-\alpha)z^2}$) yields

$$\int_{\mathbb{R}} g(z; U) z \, \phi(z) \, dz = \int_{\mathbb{R}} g'(z; U) \, \phi(z) \, dz = \mathbb{E}_Z\left[g'(Z; U)\right].$$

By definition of $g$, $g'(z; U) = \partial h(R)/\partial R_{ab}$ evaluated at the matrix with entry $(a, b)$ equal to $z$ and others fixed. Thus

$$\mathbb{E}\left[h(R) R_{ab}\right] = \mathbb{E}\left[\frac{\partial h}{\partial R_{ab}}(R)\right].$$

Stacking over all $(a, b)$ gives (27). □

**Theorem A.3** (Restate of Proposition 5.1 and 5.2). *Fix $W$ and a batch $B$. Let $A_l \in \mathbb{R}^{d_{in} \times r}$ have orthonormal columns, and let $R_l \in \mathbb{R}^{d_{out} \times r}$ have i.i.d. $\mathcal{N}(0, 1)$ entries, independently across $l$. Define $\Delta_l := R_l A_l^\top$, $\Delta := \{\Delta_l\}_{l=1}^{L}$,*

$$f_{\mu,A}(W, B) := \mathbb{E}_R f(W + \mu\Delta, B), \qquad \phi(W, \Delta; B) := \frac{f(W + \mu\Delta, B) - f(W, B)}{\mu}.$$

*Assume L-smoothness of $f(\cdot, B)$. Then for each layer $l$:*

(a)

$$\nabla_{W_l} f_{\mu,A}(W, B) A_l A_l^\top = \frac{1}{\mu} \mathbb{E}_R\left[f(W + \mu\Delta, B) R_l A_l^\top\right]. \tag{28}$$

*(b) Using $\mathbb{E}_R = 0$,*

$$\nabla_{W_l} f_{\mu,A}(W, B) \, A_l A_l^\top \;=\; \mathbb{E}_R \Big[ \phi(W, \Delta; B, B) \, R_l A_l^\top \Big]. \tag{29}$$

*(c) There exist absolute constants $c_l < \infty$ (depending only on Gaussian moments and layer shapes) such that*

$$\Big\| \nabla_{W_l} f_{\mu,A}(W, B) \, A_l A_l^\top - \big( \nabla_{W_l} f(W, B) \big) \, A_l A_l^\top \Big\|_F \;\leq\; c_l \, L \, \mu, \tag{30}$$

*Averaging over $B$ gives the population versions with $f \to F$ on both sides.*

*Proof.* **(a)** Let $h(R) := f(W + \mu\Delta, B)$ with $\Delta_l = R_l A_l^\top$. Varying $R_l$ only, the differential is

$$dh = \langle \nabla_{W_l} f(W + \mu\Delta, B), \, \mu \, dR_l A_l^\top \rangle = \mu \, \langle \nabla_{W_l} f(W + \mu\Delta, B) A_l, \, dR_l \rangle,$$

hence

$$\nabla_{R_l} h(R) = \mu \, \nabla_{W_l} f(W + \mu\Delta, B) \, A_l.$$

Note that the L-Lipschitz gradient assumption implies at most quadratic growth of $f$, which satisfies Condition (A) required by Lemma A.2. Hence we have,

$$\mathbb{E}_{R_l}[h(R) \, R_l] = \mathbb{E}_{R_l}[\nabla_{R_l} h(R)] = \mu \, \mathbb{E}_{R_l} \big[ \nabla_{W_l} f(W + \mu\Delta, B) \, A_l \big].$$

Right-multiplying by $A_l^\top$ and dividing by $\mu$,

$$\mathbb{E}_R \big[ \nabla_{W_l} f(W + \mu\Delta, B) \, A_l A_l^\top \big] = \frac{1}{\mu} \, \mathbb{E}_R \big[ f(W + \mu\Delta, B) \, R_l A_l^\top \big].$$

By (24), we have:

$$\nabla_{W_l} f_{\mu,A}(W, B) = \mathbb{E}_R \big[ \nabla_{W_l} f(W + \mu\Delta, B) \big].$$

Hence,

$$\nabla_{W_l} f_{\mu,A}(W, B) A_l A_l^T = \frac{1}{\mu} \, \mathbb{E}_R \big[ f(W + \mu\Delta, B) \, R_l A_l^\top \big].$$

which is exactly (28).

**(b)** By $\mathbb{E}_R = 0$,

$$\frac{1}{\mu} \, \mathbb{E}_R[f(W + \mu\Delta, B) \, R_l A_l^\top] = \mathbb{E}_R \Big[ \frac{f(W + \mu\Delta, B) - f(W, B)}{\mu} R_l A_l^\top \Big],$$

which yields (29).

**(c)** By the $L$-smooth descent lemma, for $h = \pm\mu\Delta$,

$$f(W + h, B) = f(W, B) + \langle \nabla f(W, B), h \rangle + R_h, \quad |R_h| \leq \frac{L}{2} \|h\|_F^2.$$

Hence

$$\phi(W, \Delta; B, B) = \langle \nabla f(W, B), \Delta \rangle + r_\mu, \qquad |r_\mu| \leq \frac{L}{2} \mu \|\Delta\|_F^2.$$

Plugging into (29),

$$\nabla_{W_l} f_{\mu,A}(W, B) \, A_l A_l^\top = \underbrace{\mathbb{E}_R \big[ \langle \nabla f(W, B), \Delta \rangle R_l A_l^\top \big]}_{(*)} + \mathbb{E}_R \big[ r_\mu R_l A_l^\top \big].$$

*Evaluate $(*)$.* Decompose layerwise:

$$\langle \nabla f(W, B), \Delta \rangle = \sum_{i=1}^L \langle \nabla_{W_i} f(W, B), R_i A_i^\top \rangle = \sum_{i=1}^L \langle \nabla_{W_i} f(W, B) A_i, \, R_i \rangle.$$

Therefore

$$(*) = \sum_{i=1}^{L} \mathbb{E}_R \big[ \langle \nabla_{W_i} f(W, B) A_i, \ R_i \rangle R_l \big] A_l^\top.$$

For $i \neq l$, independence and $\mathbb{E}[R_l] = 0$ give zero. For $i = l$, apply Lemma A.1 with $M = \nabla_{W_l} f(W, B) A_l$ and $R = R_l$ to obtain $\mathbb{E}[\langle M, R_l \rangle R_l] = M$, hence $(*) = \nabla_{W_l} f(W, B) A_l A_l^\top$.

*Bound the remainder.* Using $\|\mathbb{E}[XY]\|_F \leq \mathbb{E}[|X| \, \|Y\|_F]$ and Gaussian moment finiteness,

$$\Big\| \mathbb{E}_R \big[ r_\mu \, R_l A_l^\top \big] \Big\|_F \leq \frac{L}{2} \mu \, \mathbb{E} \big[ \|\Delta\|_F^2 \, \|R_l\|_F \big] \leq c_L \, L \, \mu.$$

for some absolute constants $c_l$. This yields (30). Averaging over $B$ proves the population statements. $\qquad \square$

### A.1.3. ISOTROPIC GAUSSIAN SMOOTHING IDENTITY FOR MEZO

**Theorem A.4.** *] Fix $W$ and a batch $B$. For each layer $l$, let $U_l \in \mathbb{R}^{d_{out} \times d_{in}}$ have i.i.d. $\mathcal{N}(0, 1)$ entries, independently across $l$, and define the full-direction block $\Delta_l := U_l$ and $\Delta := \{\Delta_l\}_{l=1}^{L}$. Define*

$$f_{\mu,\text{iso}}(W, B) := \mathbb{E}_U f(W + \mu\Delta, B), \qquad \phi(W, \Delta; B, B) := \frac{f(W + \mu\Delta, B) - f(W, B)}{\mu}.$$

*Assume $L$-smoothness of $f(\cdot, B)$. Then for each layer $l$:*

*(a)*

$$\nabla_{W_l} f_{\mu,\text{iso}}(W, B) \ = \ \frac{1}{\mu} \mathbb{E}_U \big[ f(W + \mu\Delta, B) \, U_l \big]. \tag{31}$$

*(b) By $\mathbb{E}_U = 0$,*

$$\nabla_{W_l} f_{\mu,\text{iso}}(W, B) \ = \ \mathbb{E}_U \Big[ \phi(W, \Delta; B, B) \, U_l \Big]. \tag{32}$$

*(c) There exist absolute constants $c_l < \infty$ (depending only on Gaussian moments and layer shapes) such that*

$$\big\| \nabla_{W_l} f_{\mu,\text{iso}}(W, B) - \nabla_{W_l} f(W, B) \big\|_F \ \leq \ c_L \, L \, \mu \tag{33}$$

*Averaging over $B$ yields the population versions with $f \to F$ on both sides.*

*Proof.* **(a)** By definition, we have:

$$\nabla_{W_l} f_{\mu,\text{iso}}(W, B) = \mathbb{E}_U \big[ \nabla_{W_l} f(W + \mu\Delta, B) \big].$$

Let $h(U) := f(W + \mu\Delta, B)$ with $\Delta_l = U_l$. Varying $U_l$ only,

$$dh = \big\langle \nabla_{W_l} f(W + \mu\Delta, B), \ \mu \, dU_l \big\rangle = \mu \big\langle \nabla_{W_l} f(W + \mu\Delta, B), \ dU_l \big\rangle,$$

so $\nabla_{U_l} h(U) = \mu \, \nabla_{W_l} f(W + \mu\Delta, B)$.

Applying Lemma A.2 to $U_l$ gives

$$\mathbb{E}_{U_l} \big[ h(U) \, U_l \big] = \mathbb{E}_{U_l} \big[ \nabla_{U_l} h(U) \big] = \mu \, \mathbb{E}_{U_l} \big[ \nabla_{W_l} f(W + \mu\Delta, B) \big].$$

Taking expectation over all blocks $U$ and dividing by $\mu$ yields

$$\frac{1}{\mu} \mathbb{E}_U \big[ f(W + \mu\Delta, B) \, U_l \big] = \mathbb{E}_U \big[ \nabla_{W_l} f(W + \mu\Delta, B) \big] = \nabla_{W_l} f_{\mu,\text{iso}}(W, B),$$

which is (31).

**(b)** $\mathbb{E}_U = 0$ implies

$$\frac{1}{\mu}\,\mathbb{E}_U\big[f(W + \mu\Delta, B)\,U_l\big] = \mathbb{E}_U\Big[\frac{f(W + \mu\Delta, B) - f(W, B)}{\mu}\,U_l\Big],$$

giving (32).

**(c)** By the second-order Taylor bounds , for $h = \pm\mu\Delta$,

$$f(W + h, B) = f(W, B) + \langle\nabla f(W, B), h\rangle + R_h, \quad |R_h| \leq \frac{L}{2}\|h\|_F^2.$$

Hence

$$\phi(W, \Delta; B, B) = \langle\nabla f(W, B), \Delta\rangle + r_\mu, \qquad |r_\mu| \leq \frac{L}{2}\,\mu\,\|\Delta\|_F^2.$$

Insert into (32):

$$\nabla_{W_l} f_{\mu,\text{iso}}(W, B) = \underbrace{\mathbb{E}_U\big[\langle\nabla f(W, B), \Delta\rangle\,U_l\big]}_{(*)} + \mathbb{E}_U\big[r_\mu\,U_l\big].$$

For $(*)$, decompose layerwise:

$$\langle\nabla f(W, B), \Delta\rangle = \sum_{i=1}^{L}\langle\nabla_{W_i} f(W, B), U_i\rangle.$$

Taking expectation, independence across blocks makes cross-terms vanish; for $i = l$, apply Lemma A.1 with $G = \nabla_{W_l} f(W, B)$ and $U = U_l$ to get $\mathbb{E}[\langle G, U_l\rangle U_l] = G$. Thus $(*) = \nabla_{W_l} f(W, B)$. Finally,

$$\big\|\mathbb{E}_U[r_\mu\,U_l]\big\|_F \leq \mathbb{E}|r_\mu|\,\|U_l\|_F \leq c_l\,L\,\mu.$$

by finiteness of Gaussian moments, which proves (33). Averaging over $B$ gives the population statements. $\qquad\square$

### A.1.4. CONSEQUENCES AND SPECIALIZATIONS

**AGZO.** Let $S_l^{(r)} = \text{span}(U_l^{(r)})$ be the leading activation subspace and suppose $A_l$ is an orthonormal basis that (approximately) spans $S_l^{(r)}$. By Lemma A.3,

$$\nabla_{W_l} F_{\mu,A}(W)\,A_l A_l^\top = \big(\nabla_{W_l} F(W)\big)\,A_l A_l^\top + O(L\,\mu). \tag{34}$$

In the ideal alignment case $S_l^{(r)} = \text{col}(H_l)$, using $\text{row}(\nabla_{W_l} F) \subseteq \text{col}(H_l)$ we have $\nabla_{W_l} F(W) = \nabla_{W_l} F(W)\,\Pi_{S_l^{(r)}}$, so the only bias comes from smoothing.

**MEZO.** By Lemma A.4,

$$\nabla_{W_l} F_{\mu,\text{iso}}(W) = \nabla_{W_l} F(W) + O(L\,\mu) \tag{35}$$

Hence MEZO estimates the full (isotropically smoothed) gradient, and is unbiased for $\nabla F(W)$ as $\mu \to 0$.

### A.2. Expected Cosine Similarity

#### A.2.1. EXPECTED COSINE SIMILARITY FOR AGZO

For simplification, we denote the true gradient as $G \in \mathbb{R}^{d_{out} \times d_{in}}$ and the agzo approximated gradient as $\widehat{G}$. Let $A \in \mathbb{R}^{d_{in} \times r}$ have orthonormal columns ($A^\top A = I_r$) and let $R \in \mathbb{R}^{d_{out} \times r}$ have i.i.d. $\mathcal{N}(0, 1)$ entries. In AGZO we perturb with

$$\Delta = RA^\top.$$

The AGZO estimator for layer $\ell$ can be written as

$$\widehat{\nabla}_{W_\ell}^{\text{AGZO}}(W; B) = \phi(W, \Delta(W, R); B)\,R_\ell A_\ell^\top, \tag{36}$$

where

$$\phi(W, \Delta; B) = \frac{f(W + \mu\Delta, B) - f(W, B)}{\mu}. \tag{37}$$

We assume the smoothing parameter tends to zero $\mu \to 0$, where the central difference in (37) equals the directional derivative:

$$\phi \;=\; \langle G, \Delta \rangle \;=\; \langle G, RA^\top \rangle.$$

The estimate is

$$\widehat{G}_0 \;=\; \phi\, RA^\top.$$

We use Frobenius inner product $\langle X, Y \rangle := \mathrm{tr}(X^\top Y)$ and Frobenius norm $\|X\|_F := \sqrt{\langle X, X \rangle}$. Our target is the expectation (over $R$ only)

$$\mathbb{E}_R\Big[\cos(\widehat{G}_0,\, G)\Big], \qquad \cos(\widehat{G}_0, G) := \frac{\langle \widehat{G}_0, G \rangle}{\|\widehat{G}_0\|_F\, \|G\|_F}.$$

**Theorem A.5** (Copy of Theorem 5.4). *Let $\widehat{G}_0^{\mathrm{AGZO}}$ be the noiseless AGZO estimator constructed from $\Delta = RA^\top$ with $R \sim \mathcal{N}(0, I_{d_{out} \times r})$. Then*

$$\mathbb{E}_R\big[\cos\big(\widehat{G}_0^{\mathrm{AGZO}}, G\big)\big] = \beta_{d_{out}r}\, \frac{\|GA\|_F}{\|G\|_F}, \tag{38}$$

*where*

$$\beta_D := \mathbb{E}\big[|U_1|\big], \quad U = (U_1, \ldots, U_D) \sim \mathrm{Unif}(\mathbb{S}^{D-1}), \tag{39}$$

*depends only on the product dimension $d_{out}r$. Equivalently,*

$$\beta_D = \frac{\Gamma\big(\frac{D}{2}\big)}{\sqrt{\pi}\, \Gamma\big(\frac{D+1}{2}\big)}, \tag{40}$$

*and for any $D \geq 2$, $\beta_D$ satisfies the tight bounds:*

$$\sqrt{\frac{2}{\pi D}} \;\leq\; \beta_D \;\leq\; \sqrt{\frac{2}{\pi(D-1)}}. \tag{41}$$

*Proof.* **Step 1 (numerator).** Using $\langle X, Y \rangle = \mathrm{tr}(X^\top Y)$ and cyclicity of trace,

$$\langle \widehat{G}_0,\, G \rangle = \mathrm{tr}\big((\phi RA^\top)^\top G\big) = \phi\, \mathrm{tr}\big(AR^\top G\big) = \phi\, \mathrm{tr}\big(R^\top GA\big) = \phi\, \langle R,\, GA \rangle.$$

By definition of $\phi$,

$$\phi \;=\; \langle G, RA^\top \rangle = \mathrm{tr}\big(G^\top RA^\top\big) = \mathrm{tr}\big(A^\top G^\top R\big) = \langle R, GA \rangle.$$

Hence

$$\langle \widehat{G}_0,\, G \rangle \;=\; \langle R, GA \rangle^2.$$

**Step 2 (denominator).** We have $\|\widehat{G}_0\|_F = |\phi|\, \|RA^\top\|_F$. Since $A^\top A = I_r$,

$$\|RA^\top\|_F^2 = \mathrm{tr}\big((RA^\top)^\top (RA^\top)\big) = \mathrm{tr}\big(AR^\top RA^\top\big) = \mathrm{tr}\big(R^\top RA^\top A\big) = \mathrm{tr}\big(R^\top R\big) = \|R\|_F^2.$$

Therefore $\|RA^\top\|_F = \|R\|_F$ and

$$\|\widehat{G}_0\|_F \;=\; |\phi|\, \|R\|_F \;=\; |\langle R, GA \rangle|\, \|R\|_F.$$

**Step 3 (cosine for a fixed $R$).** Combining the two steps,

$$\cos\big(\widehat{G}_0,\, G\big) = \frac{\langle \widehat{G}_0, G \rangle}{\|\widehat{G}_0\|_F\, \|G\|_F} = \frac{\langle R, GA \rangle^2}{|\langle R, GA \rangle|\, \|R\|_F\, \|G\|_F} = \frac{|\langle R, GA \rangle|}{\|R\|_F\, \|G\|_F}.$$

**Step 4 (vectorization and rotational reduction).** Let $r := \mathrm{vec}(R) \in \mathbb{R}^d$ with $d = d_{out}r$ and note $r \sim \mathcal{N}(0, I_d)$; also set $k := \mathrm{vec}(GA)$, so $\langle R, GA \rangle = r^\top k$ and $\|R\|_F = \|r\|_2$. Thus

$$\cos\big(\widehat{G}_0,\, G\big) = \frac{|r^\top k|}{\|r\|_2\, \|G\|_F} = \frac{\|k\|_2}{\|G\|_F} \cdot \frac{|r^\top \hat{k}|}{\|r\|_2}, \qquad \hat{k} := \frac{k}{\|k\|_2}.$$

By rotational invariance of $r \sim \mathcal{N}(0, I_d)$, the distribution of $\frac{r}{\|r\|_2}$ is uniform on the unit sphere $\mathbb{S}^{d-1}$. Hence

$$\mathbb{E}_R \left[ \frac{|r^\top \hat{k}|}{\|r\|_2} \right] = \mathbb{E}\big[ |U_1| \big] =: \beta_{d_{out}r},$$

where $U = (U_1, \ldots, U_{mr})$ is uniform on $\mathbb{S}^{d_{out}r-1}$. Therefore

$$\mathbb{E}_R \Big[ \cos\big(\widehat{G}_0, G\big) \Big] = \frac{\|k\|_2}{\|G\|_F} \beta_{d_{out}r} = \beta_{d_{out}r} \frac{\|GA\|_F}{\|G\|_F}.$$

**Step 5 (closed form for $\beta_{d_{out}r}$).** The marginal density of $U_1$ is

$$f_{d_{out}r}(t) = c_{d_{out}r} (1 - t^2)^{\frac{d_{out}r-3}{2}}, \quad t \in [-1, 1], \qquad c_{d_{out}r} = \frac{\Gamma(\frac{d_{out}r}{2})}{\sqrt{\pi}\,\Gamma(\frac{d_{out}r-1}{2})}.$$

Then

$$\beta_{d_{out}r} = \mathbb{E}|U_1| = 2 \int_0^1 t\, f_{d_{out}r}(t)\, dt = 2 c_{d_{out}r} \int_0^1 t\, (1 - t^2)^{\frac{d_{out}r-3}{2}}\, dt.$$

With the substitution $u = t^2$ (so $du = 2t\, dt$), we get

$$\beta_{d_{out}r} = c_{d_{out}r} \int_0^1 (1 - u)^{\frac{d_{out}r-3}{2}}\, du = c_{d_{out}r} \cdot \frac{2}{d_{out}r - 1} = \frac{\Gamma(\frac{d_{out}r}{2})}{\sqrt{\pi}\,\Gamma(\frac{d_{out}r+1}{2})}.$$

For the bound of $\beta$, please see lemma A.7. $\qquad \square$

*Remark* A.6 (Equivalent projector form). Since $A^\top A = I_r$,

$$\|GA\|_F^2 = \text{tr}(A^\top G^\top GA) = \text{tr}(G^\top G\, AA^\top) = \|G\, AA^\top\|_F^2.$$

Thus the main factor can also be written as $\|G AA^\top\|_F / \|G\|_F$, i.e. the fraction of gradient energy captured by the $r$-dimensional subspace spanned by the columns of $A$.

**Lemma A.7.** *For every integer $D \geq 2$, the sequence $\{\beta_D\}$ is strictly decreasing in $D$ and satisfies*

$$\sqrt{\frac{2}{\pi D}} \leq \beta_D \leq \sqrt{\frac{2}{\pi(D-1)}}. \tag{42}$$

*Proof.* By Gautschi's inequality (Gautschi, 1959) with $s = \frac{1}{2}$ applied at $u - \frac{1}{2}$ (so $u > \frac{1}{2}$):

$$\sqrt{u - \frac{1}{2}} < \frac{\Gamma(u + \frac{1}{2})}{\Gamma(u)}.$$

Wendel's inequality (Wendel, 1948) states that for $u > 0$ and $s \in (0, 1)$,

$$\frac{\Gamma(u + s)}{u^s\, \Gamma(u)} \leq 1.$$

Set $s = \frac{1}{2}$ and multiply by $u^{1/2}$:

$$\frac{\Gamma(u + \frac{1}{2})}{\Gamma(u)} \leq \sqrt{u} \qquad (u > 0).$$

Combining:

$$\sqrt{u - \frac{1}{2}} < \frac{\Gamma(u + \frac{1}{2})}{\Gamma(u)} < \sqrt{u}.$$

Substituting $u = \frac{D}{2}$ and multiplied by $\sqrt{\pi}$, we get:

$$\sqrt{\frac{\pi(D-1)}{2}} < \frac{1}{\beta_D} < \sqrt{\frac{\pi D}{2}}$$

By reversing we complete the proof. $\qquad \square$

### A.3. AGZO defeat MEZO in cosine similarity

We compare the noiseless expectations from theorem 5.4 and corollary 5.5:

$$\mathbb{E}_R\Big[\cos\big(\widehat{G}_0^{\text{AGZO}}, G\big)\Big] = \beta_{d_{out}r}\, \frac{\|GP_A\|_F}{\|G\|_F}, \qquad \mathbb{E}_R\Big[\cos(\widehat{G}_0^{\text{MEZO}}, G)\Big] = \beta_{d_{out}d_{in}},$$

where $A \in \mathbb{R}^{d_{in} \times r}$ is orthonormal (AGZO's subspace), $P_A = AA^\top$, and

$$\beta_D \;=\; \frac{\Gamma(\frac{D}{2})}{\sqrt{\pi}\,\Gamma(\frac{D+1}{2})} \qquad (D \in \mathbb{N}).$$

Define the *energy–capture factor*

$$\alpha(A; G) \;:=\; \frac{\|GP_A\|_F}{\|G\|_F} \in [0, 1].$$

Then

$$\mathbb{E}_R\Big[\cos(\widehat{G}_0^{\text{AGZO}}, G)\Big] \;>\; \mathbb{E}_R\Big[\cos(\widehat{G}_0^{\text{MEZO}}, G)\Big] \quad \Longleftrightarrow \quad \alpha(A; G) \;>\; \frac{\beta_{d_{out}d_{in}}}{\beta_{d_{out}r}}. \tag{43}$$

The threshold is the *exact* constant

$$\frac{\beta_{d_{out}d_{in}}}{\beta_{d_{out}r}} = \frac{\Gamma(\frac{d_{out}d_{in}}{2})}{\Gamma(\frac{d_{out}r}{2})} \cdot \frac{\Gamma(\frac{d_{out}r+1}{2})}{\Gamma(\frac{d_{out}d_{in}+1}{2})}.$$

By lemma A.7, we have

$$\frac{\beta_{d_{out}d_{in}}}{\beta_{d_{out}r}} \;<\; \frac{\sqrt{\frac{d_{out}r}{2}}}{\sqrt{\frac{d_{out}d_{in}}{2} - \frac{1}{2}}} = \frac{\sqrt{d_{out}r}}{\sqrt{d_{out}d_{in} - 1}}.$$

Hence AGZO beats MEZO if,

$$\alpha(A; G) \;>\; \frac{\sqrt{d_{out}r}}{\sqrt{d_{out}d_{in} - 1}}$$

or (by taking square)

$$\frac{\sum_{i=1}^r B_{ii}\, \sigma_i^2}{\sum_{i=1}^s B_{ii}\, \sigma_i^2} \;>\; \frac{r}{d_{in} - 1/d_{out}} \tag{44}$$

We then have the following theorem to see (44) is valid if $B$ is isotropic.

**Theorem A.8** (Copy of Theorem 5.6). *Consider a layer with gradient factorization $\nabla_{W_\ell} F(W) = Q_\ell H_\ell^\top$ and compact SVD $H_\ell = U_\ell \Sigma_\ell V_\ell^\top$ of rank $s_\ell < d_{in}$ with singular values $\{\sigma_{\ell,i}\}_{i=1}^{s_\ell}$. Let $A_\ell = U_\ell^{(r)}$ be the AGZO subspace and $B_\ell = V_\ell^\top Q_\ell^\top Q_\ell V_\ell$. If the average of its first $r$ diagonal entries is not less than the average of all diagonal entries, i.e.,*

$$\frac{1}{r}\sum_{i=1}^r B_{\ell,ii} \;\geq\; \frac{1}{s_\ell}\sum_{i=1}^{s_\ell} B_{\ell,ii}, \tag{45}$$

*and that $H_\ell$ is low-rank, i.e., $s_\ell < d_{in}$. Then*

$$\frac{\sum_{i=1}^r B_{\ell,ii}\, \sigma_{\ell,i}^2}{\sum_{i=1}^{s_\ell} B_{\ell,ii}\, \sigma_{\ell,i}^2} \;>\; \frac{r}{s_\ell} \;\geq\; \frac{r}{d_{in} - 1/d_{out}}, \tag{46}$$

*and hence, in the noiseless single-query setting,*

$$\mathbb{E}_R\big[\cos\big(\widehat{G}_0^{\text{AGZO}}, G_\ell\big)\big] \;>\; \mathbb{E}_R\big[\cos\big(\widehat{G}_0^{\text{MEZO}}, G_\ell\big)\big]. \tag{47}$$

*Moreover, when the singular values $\{\sigma_{\ell,i}\}$ are more heterogeneous (so that the leading directions carry more weighted energy), the gap in expected cosine similarity becomes larger.*

*Proof.* Here we omit the subscript $\ell$ for simplicity. Define

$$D := \frac{1}{r}\sum_{i=1}^{r} B_{ii}\sigma_i^2 - \frac{1}{s}\sum_{i=1}^{s} B_{ii}\sigma_i^2 = \frac{s-r}{rs}\sum_{i=1}^{r} B_{ii}\sigma_i^2 - \frac{1}{s}\sum_{i=r+1}^{s} B_{ii}\sigma_i^2.$$

Since $\sigma_1 > \cdots > \sigma_s \geq 0$, we have for $i \leq r$ that $\sigma_i^2 \geq \sigma_r^2$, and for $r < i \leq s$ that $\sigma_i^2 \leq \sigma_{r+1}^2$. Hence

$$D \geq \frac{s-r}{rs}\sigma_r^2\sum_{i=1}^{r} B_{ii} - \frac{1}{s}\sigma_{r+1}^2\sum_{i=r+1}^{s} B_{ii}.$$

Add and subtract $\frac{s-r}{rs}\sigma_{r+1}^2\sum_{i=1}^{r} B_{ii}$ to obtain

$$D \geq \frac{1}{rs}\left(s\sum_{i=1}^{r} B_{ii} - r\sum_{i=1}^{s} B_{ii}\right)\sigma_{r+1}^2 + \frac{s-r}{rs}\left(\sigma_r^2 - \sigma_{r+1}^2\right)\sum_{i=1}^{r} B_{ii}.$$

By the assumption,

$$s\sum_{i=1}^{r} B_{ii} - r\sum_{i=1}^{s} B_{ii} = rs\left(\frac{1}{r}\sum_{i=1}^{r} B_{ii} - \frac{1}{s}\sum_{i=1}^{s} B_{ii}\right) \geq 0.$$

Moreover, $\sigma_r^2 - \sigma_{r+1}^2 > 0$ and $B_{ii} \geq 0$. Therefore both terms on the right-hand side are nonnegative, which yields $D \geq 0$, or equivalently,

$$\frac{\sum_{i=1}^{r} B_{ii}\sigma_i^2}{\sum_{i=1}^{s} B_{ii}\sigma_i^2} > \frac{r}{s} \geq \frac{r}{d_{in} - 1/d_{out}}.$$

The second inequality is due to the fact that $s < d_{in}$ and both $s, d_{in} \in \mathbb{N}^+$. Combining with equation (44) and (43), we can conclude that AGZO generally can get gradient estimation have larger cosine similarity with the true gradient than MEZO.

$\square$

# B. Experimental Details

## B.1. Datasets

We evaluate our method on a diverse set of natural language understanding and question-answering benchmarks. The datasets used in our experiments include:

- **SST-2** (Socher et al., 2013): A single-sentence classification task focusing on sentiment analysis of movie reviews.

- **DROP** (Dua et al., 2019): A reading comprehension benchmark that requires discrete reasoning (e.g., sorting, counting, arithmetic) over paragraphs to answer questions. This dataset challenges the model's ability to perform complex logical operations beyond simple span extraction.

- **SuperGLUE Benchmark Tasks** (Wang et al., 2019): We select several challenging tasks requiring complex reasoning:

  - **BoolQ** (Clark et al., 2019): QA task with yes/no answers based on passages.
  - **MultiRC** (Khashabi et al., 2018): QA task requiring reasoning over multiple sentences.
  - **RTE**: Natural language inference task.
  - **WiC** (Pilehvar & Camacho-Collados, 2019): Word sense disambiguation task.
  - **CB** (De Marneffe et al., 2019): Few-shot multi-class entailment task.
  - **COPA** (Roemmele et al., 2011): Causal reasoning task.

- **SQuAD** (Rajpurkar et al., 2016): A standard reading comprehension dataset based on Wikipedia articles.

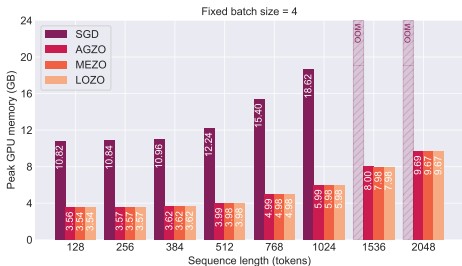
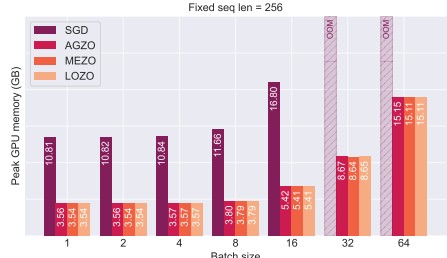

*(a)* Fixed batch size $= 4$, varying sequence length.      *(b)* Fixed sequence length $= 256$, varying batch size.

*Figure 4.* Peak GPU memory usage when fine-tuning Pangu-1B on DROP.

## B.2. Implementation and Hyperparameters

All zeroth-order optimization methods (AGZO, MeZO, and LOZO) are implemented using the same codebase to ensure a fair comparison. For all ZO experiments, we perform fine-tuning for a total of 20,000 steps. This fixed budget allows us to directly compare the convergence speed and final performance of different estimators under identical computational constraints. We set the smoothing parameter (perturbation scale) $\mu = 10^{-7}$ for all methods (AGZO, MeZO, and LOZO) on Qwen3 models and $\mu = 10^{-4}$ on the Pangu model. This value was chosen to minimize discretization error while maintaining numerical stability.

For AGZO, we set the subspace rank $r_\ell = 1$ for all linear layers, a design choice driven by both signal quality and memory efficiency. As shown in our spectral analysis (Figure 1), the singular values of activation matrices decay rapidly; thus, a rank-1 setting concentrates the finite-difference perturbation along the single most dominant direction of the activation landscape. This maximizes the signal-to-noise ratio of the gradient estimate by avoiding the exploration of low-energy directions that contribute little to the true gradient. Furthermore, this rank-1 basis minimizes the memory overhead for storing the subspace information ($A_\ell$), ensuring the peak memory footprint remains nearly identical to that of standard inference.

## B.3. Additional Experiments on Pangu-1B

In this section, we provide supplementary experimental details for the `openPangu-embedded-1B` model (Chen et al., 2025a). While the main text (Section 6) reports the standard GPU fine-tuning performance, here we focus on (1) specific implementation details regarding low-precision (BF16) optimization, (2) cross-platform verification on Ascend NPUs, and (3) memory footprint analysis specific to this architecture.

**Precision and Hyperparameters (BF16).** Unlike the Qwen experiments where models are loaded in FP32, we conduct Pangu experiments using BF16 precision to simulate resource-constrained edge scenarios. Since FP32 has significantly higher mantissa precision than BF16, zeroth-order optimization requires a larger perturbation magnitude to overcome numerical noise. Accordingly, we adjust the perturbation parameter to $\mu = 1 \times 10^{-4}$ for Pangu (compared to $10^{-7}$ for FP32 models).

**Peak Memory Footprint on Pangu-1B.** We empirically validate the memory efficiency of AGZO on the Pangu architecture by measuring the peak GPU memory footprint during fine-tuning on the DROP task with Testbed One. As shown in Figure 4a, when fixing the batch size at 4 and scaling the sequence length, the First-Order (FO) baseline encounters Out-Of-Memory (OOM) errors at a sequence length of 1,536. In contrast, AGZO maintains a strictly linear scaling, consuming only 9.69 GB even at a context length of 2,048. Similarly, with a fixed sequence length of 256 (Figure 4b), SGD triggers OOM at a batch size of 32, whereas AGZO remains operative up to a batch size of 64. This confirms that the memory advantages of AGZO observed on Qwen models (Section 6.4) generalize to different architectures, with the activation-guided subspace construction introducing negligible memory overhead.

## B.4. Additional Experiments on Qwen3-8B and OpenMathReasoning

To further evaluate whether the gains of AGZO persist beyond the model scales and tasks reported in the main text, we conduct additional experiments on Qwen3-8B. We also include OpenMathReasoning as a reasoning-oriented benchmark to

*Table 5.* Additional Qwen3-8B results.

| Method | OpenMath | SST-2 | COPA | CB |
|--------|----------|-------|------|-----|
| AGZO | **0.640** | **0.932** | **0.820** | **0.928** |
| MeZO | 0.560 | 0.916 | 0.800 | 0.910 |
| LOZO | 0.580 | 0.907 | 0.790 | 0.910 |

*Table 6.* Comparison with LoRA on Qwen3-0.6B.

| Method | SST-2 | COPA | CB | BoolQ | MultiRC |
|--------|-------|------|-----|-------|---------|
| AGZO | **0.877** | 0.740 | **0.892** | **0.724** | 0.756 |
| LoRA | 0.834 | **0.740** | 0.817 | 0.684 | **0.771** |

*Table 7.* Throughput comparison on Qwen3-0.6B, measured in steps per second.

| Method | SST-2 | COPA | MultiRC | SQuAD | WiC |
|--------|-------|------|---------|-------|-----|
| AGZO | 1.328 | 1.517 | 0.386 | 0.733 | 1.373 |
| LOZO | 1.488 | 2.054 | 0.505 | 0.783 | 1.386 |
| MeZO | 1.238 | 2.302 | 0.497 | 0.775 | 1.455 |

*Table 8.* Cosine-similarity diagnostic on GPT2-small / SST-2.

| Method | Cosine similarity |
|--------|-------------------|
| AGZO, exact SVD | 0.0124 |
| AGZO, power iteration $K = 1$ | 0.0082 |
| AGZO, power iteration $K = 3$ | 0.0123 |
| AGZO, power iteration $K = 5$ | 0.0124 |
| MeZO | 0.0015 |
| LOZO | 0.0014 |

complement the classification and QA tasks considered in the main experiments. The results are reported in Table 5.

The results show that AGZO continues to improve over MeZO and LOZO on Qwen3-8B. The improvement on OpenMath-Reasoning suggests that the benefit of activation-guided perturbations is not limited to lightweight classification tasks. At the same time, these results should be interpreted as additional evidence at a larger model scale rather than an exhaustive evaluation of all large-model regimes.

These results provide additional evidence that the gains of AGZO are not specific to Qwen3 or Pangu models. AGZO improves over both dense isotropic perturbations and data-independent low-rank perturbations on this additional model family.

### B.5. Comparison with LoRA

We additionally compare AGZO with LoRA on Qwen3-0.6B. LoRA and ZO fine-tuning represent two different memory-efficient adaptation paradigms. LoRA reduces the number of trainable parameters by introducing low-rank adapter modules and trains these adapters with backpropagation. In contrast, ZO fine-tuning avoids backpropagation and updates the original model parameters using only forward evaluations. Therefore, LoRA and AGZO should be viewed as parallel and potentially complementary approaches, rather than methods that directly replace each other.

Table 6 reports the comparison. This experiment is intended to provide practical context, not to claim that AGZO universally dominates parameter-efficient fine-tuning methods.

AGZO is competitive with LoRA in this setting, obtaining stronger results on SST-2, CB, and BoolQ, matching LoRA on COPA, and underperforming LoRA on MultiRC. These results suggest that activation-guided ZO fine-tuning can provide a viable forward-only alternative in memory-constrained settings, while LoRA remains an important and complementary parameter-efficient fine-tuning approach.

### B.6. Runtime and Throughput

AGZO introduces additional computation because it extracts activation-informed subspaces during the forward pass. This overhead mainly comes from power iteration and orthonormalization for the activation matrices. To quantify the practical cost, we report throughput in steps per second on Qwen3-0.6B in Table 7.

The results show that AGZO's throughput remains in the same practical regime as the other ZO baselines. Thus, AGZO should be understood as trading a moderate amount of additional computation for improved update quality while retaining the forward-only memory profile.

*Table 9.* Rank ablation on Qwen3-0.6B / SST-2.

| Rank $r$ | 1 | 4 | 16 |
|---|---|---|---|
| Test accuracy | **0.877** | 0.870 | 0.863 |

### B.7. Rank Ablation

The main experiments use rank $r = 1$ for all linear layers. This choice may seem conservative because higher-rank activation subspaces can capture more projected gradient energy. However, actual ZO optimization does not directly use the projection of the true gradient. Instead, it samples a stochastic perturbation direction inside the chosen subspace. Increasing the rank enlarges the activation subspace and may capture more gradient energy, but it also increases the effective sampling dimension of the single-query estimator, which can weaken the instantaneous directional quality of a sampled perturbation.

To examine this trade-off, we conduct a rank ablation on Qwen3-0.6B fine-tuned on SST-2. The results are reported in Table 9.

The results support $r = 1$ as an effective operating point in our single-query setting. It concentrates exploration along the dominant activation direction while minimizing memory and computational overhead. Adaptive layer-wise rank selection remains an interesting direction for future work.

### B.8. Power-Iteration Ablation and Low-Rank Baseline Diagnostic

AGZO uses power iteration to approximate the leading activation subspace. We evaluate the effect of the number of power-iteration steps $K$ and compare against exact SVD on GPT2-small / SST-2. We also include MeZO and LOZO in the same diagnostic to distinguish activation-guided low-rank perturbations from data-independent low-rank perturbations.

The diagnostic shows that $K = 3$ closely matches the exact-SVD version, while $K = 1$ is less accurate. The comparison with LOZO also indicates that the gain is not merely due to imposing a low-rank perturbation structure. LOZO is low-rank but data-independent, whereas AGZO uses the activation-informed subspace. This supports the interpretation that activation guidance is the key source of the improved directional fidelity.

