# OpenReview forum: "AGZO: Activation-Guided Zeroth-Order Optimization for LLM Fine-Tuning"
_ICML.cc/2026/Conference — ICML 2026 regular_

### Official Review · Reviewer_T243 · 2026-03-10

**Soundness:** 3
**Presentation:** 3
**Significance:** 3
**Originality:** 3
**Overall Recommendation:** 5
**Confidence:** 4

**Summary:**

This paper focuses on the concept of improving memory-efficient fine-tuning of large language models through better zeroth-order update directions. They attempt to analyze an important concept: whether the gradient information relevant for updating large linear layers is concentrated in a low-dimensional subspace revealed by the model’s forward-pass activations. Building on the observation that, for a linear layer, the gradient factorizes through the activation matrix and is therefore confined to the activation-induced subspace, the paper proposes AGZO, which extracts a compact activation basis on the fly and restricts zeroth-order perturbations to that basis for linear layers, while preserving the forward-only, low-memory character of prior methods. The paper then supports this design through a theoretical view of AGZO as optimizing a subspace-smoothed objective, a cosine-similarity result showing improved directional alignment over dense isotropic perturbations under stated conditions, and experiments on Qwen3 and Pangu models showing better gradient alignment, generally stronger downstream results than MeZO and LOZO, and peak memory close to other forward-only baselines.

**Compliance With Llm Reviewing Policy:**

Affirmed.

**Key Questions For Authors:**

- Why is the gradient-alignment experiment only reported against MeZO, not also against LOZO? Since LOZO is the main structured baseline, the strongest mechanistic comparison would be to show whether AGZO’s advantage in gradient cosine remains clear against LOZO, not only against dense isotropic perturbations.

- How much of AGZO’s gain over prior methods comes from using an activation-informed subspace, as opposed to simply restricting the search to a low-rank space?

- The paper makes a convincing case that AGZO preserves peak memory relative to MeZO and LOZO, but it also adds a subspace-extraction step during each forward pass. How should readers think about the memory claim relative to the extra computation introduced by power iteration?

- It would help to state more explicitly whether AGZO is meant as a generally applicable ZO recipe, or as a method whose principled core is specific to architectures dominated by linear transformations.

**Limitations:**

No, the paper should discuss limitations more directly. In particular, it should clarify that the theory is narrower than the full empirical comparison, that the method depends on specific choices such as linear-layer guidance and r=1, and that the practical tradeoff is only partially characterized because memory is measured carefully but runtime overhead is not, which makes the impact statement limited.

**Strengths And Weaknesses:**

Soundness:

- The theoretical section is reasonably well matched to the method. The paper does not only claim that AGZO works; it formalizes AGZO as a projected estimator of a subspace-smoothed objective and gives conditions under which it should have better gradient alignment than MeZO.
- The empirical section checks the mechanism directly, not only end-task accuracy. In particular, the gradient-alignment experiment is well chosen because it tests the paper’s main conceptual claim rather than relying purely on downstream results.
- The theoretical comparison is strongest against isotropic MeZO, not against LOZO, which is the most relevant structured low-rank baseline in the experiments. So the theory justifies the direction of the method, but does not fully establish superiority over all practical alternatives considered in the paper.

Presentation:

- The narrative is coherent across sections. The same core claim introduced early, namely that activations reveal useful gradient structure, is revisited in the method, theory, and experiments.
- The paper is still somewhat harder to read than necessary for a general ML audience. The core idea is intuitive, but the exposition sometimes leans heavily on optimization and subspace language before making the intuition plain.

Significance:

- The contribution is practically meaningful because it aims to improve optimization quality without giving up the main appeal of forward-only ZO methods. The memory study supports the claim that AGZO preserves the same basic memory regime as MeZO and LOZO while remaining far below first-order training.
- The significance is strongest within the niche of memory-efficient ZO fine-tuning, rather than broadly across optimization or LLM training as a whole. This feels like a solid advance in an active subarea, not a result that changes the broader landscape of model training.
- Practical impact is somewhat under-characterized because the paper focuses much more on memory than on full training efficiency. For a method motivated by resource constrained deployment, runtime and throughput would matter to the overall significance.

Originality:

- The contribution goes beyond a minor engineering tweak because it offers a new design principle for this line of methods: perturbations should depend on current internal model structure, not only on parameter dimensions.

---

> ### Author Rebuttal · Authors · 2026-03-30
>
> We thank the reviewer for the valuable comments and suggestions. We are very pleased to have received positive feedback from the reviewer. Below we respond point by point.
>
> **Q1(Gradient-alignment experiment ):**
>
> Figure 2 compares AGZO against MeZO because MeZO is the baseline that directly matches the isotropic perturbation model analyzed in our theory, so this gives the cleanest theory-to-experiment comparison. That said, the reviewer is right that LOZO is also an important empirical baseline.
>
>  LOZO uses low-rank but still data-independent random perturbations. It does not have the mechanism that would systematically bias perturbations toward the true gradient. To address this point directly, we additionally measured cosine similarity on GPT2-small / SST-2:
>
> |                | AGZO(SVD) | AGZO(Power Iteration K=1) | AGZO(Power Iteration K=3) | AGZO(Power Iteration K=5) | MeZO   | LOZO   |
> | -------------- | --------- | ------------------------- | ------------------------- | ------------------------- | ------ | ------ |
> | Cos similarity | 0.0124    | 0.0082                    | 0.0123                    | 0.0124                    | 0.0015 | 0.0014 |
>
> These results show that the improvement comes from activation-guided subspace selection, rather than merely imposing a random low-rank structure. They also show that the default power-iteration approximation already closely matches the exact-SVD version in this diagnostic. We will include a more detailed analysis in the revised version and update Figure 2 to include LOZO as well.
>
> **Q2 (AGZO’s gain from using an activation-informed subspace):**
>
> LOZO is precisely the baseline corresponding to “simply restricting the perturbation to a low-rank space”. It uses low-rank but still data-independent random perturbations. AGZO differs in the critical additional ingredient that the low-rank subspace is activation-informed. Empirically, Tables 1–3 show that AGZO consistently outperforms LOZO across the reported settings, which indicates that low-rank restriction by itself is not sufficient to explain the gain. In addition, as shown in our response to Q1, AGZO achieves much higher cosine similarity than both MeZO and LOZO.
>
> **Q3 (extra computation of AGZO):**
>
> To characterize this tradeoff more explicitly, we additionally report throughput (steps/sec) on Qwen3-0.6B:
>
> |      | SST2  | COPA  | MultiRC | Squad | WIC   |
> | ---- | ----- | ----- | ------- | ----- | ----- |
> | AGZO | 1.328 | 1.517 | 0.386   | 0.733 | 1.373 |
> | LOZO | 1.488 | 2.054 | 0.505   | 0.783 | 1.386 |
> | MEZO | 1.238 | 2.302 | 0.497   | 0.775 | 1.455 |
>
> These numbers show that AGZO does introduce additional computation relative to MeZO/LOZO, which is expected. At the same time, the overhead remains in the same practical throughput regime rather than changing the computational order of the method. This tradeoff is in fact aligned with the basic motivation of ZO methods: using more computation time to reduce memory cost. AGZO follows the same logic. It adds some extra computation, but keeps the strict memory efficiency of forward-only ZO, while yielding substantially stronger downstream performance than prior ZO baselines.
>
> **Q4 (Scope of AGZO):**
>
> We thank the reviewer for this suggestion and agree that the scope should be stated more explicitly. AGZO is not intended as an equally principled recipe for every architecture. Rather, its core motivation is strongest for linear-dominated architectures, which already cover the main regime of modern Transformer-style LLMs. In these models, most trainable parameters are associated with linear transformations, such as those in attention and MLP blocks.

---

> > ### Author Rebuttal · Reviewer_T243 · 2026-04-02
> >
> > Thank you for the rebuttal. The additional LOZO cosine-similarity result, the clarification that AGZO is primarily motivated for linear-dominated architectures, and the new throughput numbers address my main concerns and strengthen the paper. I still recommend slightly moderating claims such as “consistently outperforms,” but overall the rebuttal improves my confidence in the submission and does not change my positive assessment.

---

> > > ### Author Response · Authors · 2026-04-08
> > >
> > > Thank you for the thoughtful and positive assessment. We are glad that our rebuttal addressed your main concerns. in the revision, we will also moderate the claims and further clarify the scope and limitations.

---

### Official Review · Reviewer_GpVD · 2026-03-11

**Soundness:** 3
**Presentation:** 3
**Significance:** 2
**Originality:** 2
**Overall Recommendation:** 3
**Confidence:** 5

**Summary:**

This paper proposes **AGZO (Activation-Guided Zeroth-Order Optimization)** for LLM fine-tuning under memory-constrained settings. The key difference from prior ZO methods is that AGZO first extracts a low-dimensional subspace from the activations of each minibatch, and then applies random perturbations only within that activation-informed subspace. In other words, it does not eliminate randomness, but rather constrains random perturbations to directions that are expected to be more relevant to the true gradient. The paper evaluates the method on relatively small Qwen3 and Pangu models (up to 4B parameters) and a limited set of NLU/QA tasks, and reports that AGZO generally outperforms MeZO and LOZO. It also claims to maintain a peak GPU memory footprint comparable to other forward-only ZO baselines, while achieving relatively higher gradient cosine similarity than MeZO.

However, the evaluation leaves important gaps from a practical perspective. The paper does not compare AGZO against strong memory-efficient fine-tuning baselines such as **LoRA** or **QLoRA**, making it difficult to assess how competitive AGZO is in terms of actual memory savings or downstream performance. It also does not compare against FO-based memory-saving techniques such as **activation checkpointing**. Furthermore, although AGZO performs subspace extraction at every batch and every linear layer, the paper does not report the associated computational overhead in terms of **runtime, latency, throughput, or step time**. Therefore, while the paper is meaningful as an improvement within the ZO family, it does not yet convincingly establish AGZO as a practically competitive alternative to broader memory-efficient fine-tuning baselines.

**Compliance With Llm Reviewing Policy:**

Affirmed.

**Final Justification:**

My final recommendation remains Weak Reject. I appreciate the paper’s clear and original core idea—using activation-informed subspaces to guide zeroth-order perturbations—and I agree that the method is a meaningful improvement over prior ZO baselines within the scope of the presented experiments. The rebuttal was helpful and improved my evaluation: the added discussion on throughput, rank/power-iteration ablations, and results on a larger model / reasoning-oriented task strengthen the empirical case, and the clarification about perturbation resampling resolves a possible misunderstanding about how randomness is used during training. However, my main concerns were only partially addressed. In particular, I still find that the evidence more strongly supports relative improvement over prior ZO methods than the stronger narrative of absolute gradient alignment or a “high-precision” estimator, and the paper still does not fully establish broader practical competitiveness against stronger memory-efficient fine-tuning baselines or sufficiently address robustness and evaluation breadth. Overall, I view the work as technically interesting, reasonably sound, and clearly presented, with good originality, but its significance remains somewhat limited by the current experimental scope and by the remaining gap between the paper’s strongest claims and the evidence provided.

**Key Questions For Authors:**

1. **How should readers interpret the very small cosine values in Figure 2?** If the empirical cosine similarity between estimated and true gradients is only on the order of 10^{-4}, then the evidence supports only relative superiority over MeZO, not strong absolute alignment. (Weakness 1)
2. **Does Section 6.1 really justify the stronger narrative of “alignment to the true gradient” or a “high-precision” estimator?** Given that the reported cosine similarities remain extremely small in absolute terms, it seems more accurate to describe AGZO as relatively better than MeZO, rather than strongly aligned with the true gradient. (Weakness 2)
3. **How stable are the results across random seeds?** The method regenerates perturbations from sampled seeds, so seed variance is an inherent concern. Yet the paper does not appear to report multi-seed averages or standard deviations. (Weakness 3)
4. The reported tasks are mostly small NLU/QA datasets. Can the authors show that the claimed gains persist on broader, more demanding evaluations such as MMLU and commonsense reasoning tasks? (Weakness 4)
5. **Please compare AGZO against strong practical baselines such as LoRA, QLoRA, and activation checkpointing in terms of both memory usage and computational cost.** Since these are among the most established approaches for memory-efficient fine-tuning, omitting them leaves the practical significance of AGZO insufficiently validated. (Weakness 5)
6. **What is the actual computational overhead of AGZO?** The paper uses power iteration at every linear layer and every batch, with QR and matrix products inside the loop. What are the wall-clock step time, tokens/sec, and throughput relative to MeZO and LOZO? (Weakness 6)
7. **How sensitive is AGZO to the number of power-iteration steps and the chosen rank?** The paper fixes a small power-iteration count and uses rank 1 for all layers, but does not show ablations over either dimension. (Weakness 7)
8. **How sensitive is AGZO to minibatch heterogeneity?** Since AGZO extracts a single minibatch-level subspace for each layer, its perturbation space may be dominated by the largest activation modes in the batch. To test this directly, please compare **homogeneous batching** and **intentionally heterogeneous batching** under the same training data, token budget, optimization steps, and evaluation protocol. A multitask mixture such as Super-NaturalInstructions would be particularly suitable for this analysis. (Weakness 8)
9. **How well does AGZO scale beyond the relatively small models evaluated in the paper?** Since all experiments are conducted on models up to 4B parameters, it remains unclear whether the claimed benefits would hold for larger-scale LLMs, where the per-layer subspace extraction overhead may become more significant. (Weakness 9)
10. **Why is the FO comparison budget-mismatched?** The paper states that ZO methods are trained for 20,000 steps, while FO is trained for 1,000 steps. This makes the AGZO-vs-FO interpretation much harder to assess. (Weakness 10)

**Limitations:**

yes

**Strengths And Weaknesses:**

# Strengths

- The paper introduces a clear and principled idea: instead of using fully data-independent perturbations, AGZO constrains random perturbations to activation-informed low-dimensional subspaces. This makes the method more structurally motivated and more data-aware than prior ZO baselines such as MeZO and LOZO.
- Within the reported experimental setup, AGZO generally improves over prior ZO baselines such as MeZO and LOZO. This is enough to support the narrower claim that activation-guided perturbations are a useful refinement of existing ZO estimators.
- The method also provides a conceptually meaningful step beyond prior ZO designs: its contribution is not removing randomness, but constraining randomness to directions that are expected to be more relevant to the true gradient. This is a cleaner and more structured design than isotropic perturbations or purely random low-rank perturbations.

# Weaknesses

1. Figure 1 only demonstrates that the activation subspace provides a plausible support for the true gradient. However, Figure 2 shows that the actual AGZO estimator remains very weak in absolute directional fidelity. In other words, the method may identify where the gradient lives, but still performs poorly at estimating the gradient direction itself.
2. Section 6.1 overstates the empirical evidence. While AGZO consistently achieves higher cosine similarity than MeZO, the reported alignment remains extremely weak in absolute terms, so the experiment does not justify the stronger narrative of “alignment to the true gradient” or a “high-precision” estimator.
3. AGZO remains seed-dependent, but the paper does not report multi-seed statistics or robustness analyses, leaving the stability of the reported gains unclear.
4. The evaluation is narrow, focusing on r**elatively small models** and a **limited set of NLU/QA tasks**, which makes it difficult to assess broader generality. The paper does not evaluate on broader reasoning-heavy benchmarks(MMLU and Commonsense reasoning, etc.), and **even reports fewer tasks for the larger Qwen3-4B setup than for Qwen3-0.6B**.
5. The paper does not compare against strong practical memory-efficient fine-tuning baselines such as LoRA, QLoRA, or activation checkpointing. Without these comparisons, it is difficult to assess AGZO’s real practical competitiveness in terms of memory savings or downstream performance.
6. Although AGZO performs per-batch, per-layer subspace extraction, **the paper does not report runtime, latency, throughput, or step-time overhead**, leaving its practical efficiency relative to other ZO algorithms and PEFT methods unclear. This is especially important because the method introduces repeated power-iteration and QR/GEMM operations at every linear layer.
7. **The hyperparameter selection is also underdeveloped.** The paper fixes a small power-iteration count(K) and uses rank-1(r) subspaces for all linear layers, but does not provide sensitivity analyses over the number of power-iteration steps or the rank choice, nor does it quantify the latency tradeoff induced by those settings.
8. **AGZO may struggle with heterogeneous minibatches.** By extracting only a single subspace per layer from the whole minibatch, it may be overly influenced by dominant activation patterns and fail to capture other relevant modes. This concern is strengthened by the paper’s universal rank-1 design.
9. The paper does not convincingly address scalability, as all experiments are conducted on relatively small models (up to 4B), leaving its effectiveness on larger-scale LLMs unclear.
10. The comparison to FO is not fully fair, as ZO and FO are trained with different numbers of optimization steps, making the reported gap difficult to interpret.

---

> ### Author Rebuttal · Authors · 2026-03-31
>
> We thank the reviewer for the constructive comments. Below is the detailed rebuttal.
>
> **Q1 and Q2:**
>
> A high value in Figure 1 means that the true gradient largely lies in that subspace. However, actual ZO optimization does not directly use the subspace projection. Instead, it samples a single random direction within the corresponding subspace and forms a stochastic estimator. Therefore, even if the true gradient is well contained in a subspace, a single sampled perturbation within that subspace may still be only weakly aligned with the true gradient. This is also consistent with our theory in Sec. 5.2. Figure 2 shows that AGZO consistently improves directional fidelity over MeZO. More importantly, this relative gain also translates into improved downstream fine-tuning performance in our end-to-end experiments. In the revision, we will adjust the wording accordingly to avoid overclaiming.
>
> **Q3**:
>
> The randomness of AGZO is not tied to one fixed seed throughout training. The perturbations are resampled at every iteration from fresh random seeds. The seed is only used to regenerate the same perturbation within that iteration when forming the update. Therefore, the reported performance is already obtained under repeated stochastic perturbations across the entire optimization trajectory, rather than depending on a single fixed random draw.
>
> **Q4 and Q9:**
>
> To directly respond to these concerns, we added a new experiment on Qwen3-8B together with OpenMathReasoning, since the suggested MMLU are primarily test benchmarks and less suitable for the SFT. The results are as follows:
>
> |          | AGZO  | MeZO  | LOZO  |
> | -------- | ----- | ----- | ----- |
> | OpenmathReasoning | 0.640 | 0.560 | 0.580 |
> | SST2     | 0.932 | 0.916 | 0.907 |
> | COPA     | 0.820 | 0.800 | 0.790 |
> | CB       | 0.928 | 0.910 | 0.910 |
>
> These results show that AGZO is with consistent gains on a larger model as well as on a reasoning task. This is consistent with our theoretical analysis, which is not based on  assumptions tied to a specific model size or task.
>
> **Q5:**
>
> LoRA and AGZO are not mutually exclusive baselines, but operate on different design axes. LoRA constrains the trainable parameterization, whereas AGZO can enable full parameter finetuning. To provide additional practical context, we also compared AGZO against LoRA on Qwen3-0.6B:
>
> |      | SST2  | COPA  | CB    | BoolQ | MultiRC |
> | ---- | ----- | ----- | ----- | ----- | ------- |
> | AGZO | 0.877 | 0.740 | 0.892 | 0.724 | 0.756   |
> | LoRA | 0.834 | 0.740 | 0.817 | 0.684 | 0.771   |
>
> The results show that AGZO is generally better than LoRA. We will add a more complete comparison in the revision.
>
> **Q6:**
>
> We now report throughput in terms of steps/sec for Qwen3-0.6B on some datasets.
>
> |      | SST2  | COPA  | MultiRC | Squad | WIC   |
> | ---- | ----- | ----- | ------- | ----- | ----- |
> | AGZO | 1.328 | 1.517 | 0.386   | 0.733 | 1.373 |
> | LOZO | 1.488 | 2.054 | 0.505   | 0.783 | 1.386 |
> | MEZO | 1.238 | 2.302 | 0.497   | 0.775 | 1.455 |
>
> These numbers show that the overhead of AGZO remains in the same practical throughput regime rather than changing the computational order of the method.
>
> **Q7:**
>
> For rank $r$, we include the following ablation on Qwen3-0.6B / SST-2. Due to space limits, please see our response to reviewer whtE (Q1) for a detailed illustration.
>
> | Rank     | 1     | 4     | 16    |
> | -------- | ----- | ----- | ----- |
> | Test Acc | 0.877 | 0.870 | 0.863 |
>
> For the power-iteration step, we evaluate this on GPT2-small / SST-2:
>
> |                | AGZO(SVD) | AGZO(Power Iteration K=1) | AGZO(Power Iteration K=3) | AGZO(Power Iteration K=5) | MeZO   | LOZO   |
> | -------------- | --------- | ------------------------- | ------------------------- | ------------------------- | ------ | ------ |
> | Cos similarity | 0.0124    | 0.0082                    | 0.0123                    | 0.0124                    | 0.0015 | 0.0014 |
>
> These results support our choices of $r=1$ and $K=3$ as practical settings for AGZO. we will include a more thorough ablation study in the revision.
>
> **Q8:**
>
> Our experiments show tht AGZO consistently improves over prior ZO baselines on commonly used benchmarks. This suggests that, under the normal training setups studied in the paper, minibatch-level activation variation does not prevent AGZO from extracting useful perturbation subspaces in practice. The reviewer’s proposed setting with deliberately heterogeneous batching is interesting, but it targets a broader scenario than the one studied in the current submission.
>
> **Q10:**
>
> FO is included primarily as a reference upper bound, not as the main controlled comparator to ZO. Since FO uses exact backpropagation gradients, it converges in far fewer steps than ZO. In our setup, 1000 steps was already sufficient for FO to reach convergence. This evaluation protocol is also consistent with common practice in prior ZO fine-tuning work.

---

> > ### Author Rebuttal · Reviewer_GpVD · 2026-04-03
> >
> > After reading the rebuttal, I appreciate that the authors addressed several of my concerns by adding **results on a larger model**, **a reasoning-oriented task**, **a LoRA comparison**, **throughput numbers**, and **an ablation on rank/power iteration**. These additions strengthen the paper and clarify that **AGZO is a meaningful improvement within the ZO fine-tuning setting**.
> >
> > However, I still have some remaining reservations about **practical competitiveness** and **robustness**. In particular, the rebuttal does not yet fully address **variance across independent runs**, and the additional practical comparisons remain limited relative to stronger memory-efficient alternatives such as **QLoRA** or **FO methods with memory-saving techniques**. I am also still not fully convinced that the current evidence is sufficient to support **broader claims beyond the ZO family**, especially since the **absolute gradient-alignment signal remains weak** and some of the new experiments are still relatively narrow. Overall, while the rebuttal improves my assessment and resolves enough concerns to move from **reject to weak reject**, I believe the paper would still benefit from a **more comprehensive practical evaluation** in the final version.

---

> > > ### Author Response · Authors · 2026-04-08
> > >
> > > Thank you for the thoughtful follow-up. We are encouraged that the reviewer found the additional experiments helpful and that they clarified AGZO’s contribution within the ZO fine-tuning setting. We are also glad that our clarification on perturbation resampling helped resolve a possible misunderstanding about how randomness is used during training.
> > >
> > > To address the robustness concern more directly, we have now run AGZO on SST-2 with 10 independent seeds (see [link](https://anonymous.4open.science/r/ICMLrebuttal8834/agzo_sst2_results.png)). The mean/std trajectories show stable optimization behavior and consistent test performance across runs.
> > >
> > > On QLoRA and FO methods with memory-saving techniques such as activation checkpointing, we respectfully view them as valuable practical references, but not the most direct like-for-like baselines for the core question studied here. Our paper focuses on improving update quality within the forward-only, full-parameter ZO fine-tuning regime. This is also the scope emphasized in the paper itself. We will make this scope more explicit in the revision.

---

### Official Review · Reviewer_q6JG · 2026-03-12

**Soundness:** 2
**Presentation:** 3
**Significance:** 3
**Originality:** 3
**Overall Recommendation:** 4
**Confidence:** 5

**Summary:**

Authors provide a novel method that utilizes activation information for the gradient approximation. They provide experimental results and theoretical justification.

**Compliance With Llm Reviewing Policy:**

Affirmed.

**Final Justification:**

After rebuttal change to Weak Accept

**Key Questions For Authors:**

Above in Strengths And Weaknesses

**Limitations:**

Not enough elaborated in text.

**Strengths And Weaknesses:**

Although the paper is sound and claims are justified, there are several concerns:

1. FO is always worse in your measurements. This contradicts with the overall idea that gradient-based methods are better than ZO, just because the latter are only approximating the former.
2. Figure 2. Although the chart shows intuition that new gradients are better aligned with true ones, the scale of the chart is so small - so it is still around 0.
3. Fine-tuning Qwen-0.6B on such hardware can hardly result in OOM.
4. Variety and size of models. Models of bigger size, at least 8B, 13B, 70B should be covered. MoE and Dense models. Multimodal models. Otherwise, it is not clear how well this method works for models that are now defining the demanded landscape both for science and industry.

Also, there are some typos and structuring issues.
I like the idea, but the paper should be considerably improved. I can reconsider your paper if you bring bigger models and show your superiority there.

---

> ### Author Rebuttal · Authors · 2026-03-31
>
> We thank the reviewer for the constructive comments. We respond point by point below.
>
> **W1(FO is always worse):**
>
> We respectfully believe there may be a misunderstanding here. Our paper does not claim that FO is worse than ZO. On the contrary, the paper consistently presents FO as the stronger reference whenever it is feasible to run under the memory budget. Our claim is that AGZO improves over prior ZO baselines and narrows the gap to FO, not that it surpasses FO. In Table 1 and 2, the captions explicitly state “Bold: ZO’s best results”. This is entirely consistent with the expected hierarchy that FO uses exact backpropagation gradients, whereas ZO relies on stochastic gradient estimation. To avoid any ambiguity, we will further clarify this wording in the revision and make the intended comparison to FO more explicit.
>
> **W2 (Interpretation of Figure 2):**
>
> Figure 2 is not intended to show that AGZO nearly recovers the full FO gradient. Rather, it is intended to show that, under the same forward-only ZO budget, AGZO yields a better-aligned stochastic estimator than random baseline. More importantly, this relative gain is not merely geometric: it also translates into improved downstream fine-tuning performance in our end-to-end experiments.
>
> In high-dimensional random-direction ZO, very small absolute cosine values are expected even for meaningful estimators, because the estimator is still formed from a single sampled perturbation direction. This is also consistent with our theory. In Section 5.2, the expected cosine contains a dimension-dependent factor $\beta_D$, with bounds scaling on the order of $1/\sqrt{D}$. Thus, when the effective perturbation dimension is extremely large, absolute cosine values are naturally small.
>
> **W3 (Fine-tuning Qwen-0.6B can hardly result in OOM):**
>
> We would like to clarify this point more precisely. For full-parameter FP32 FO-SGD, a standard layerwise memory accounting is $M_{\mathrm{FO}} \approx |x|+\sum_{\ell=1}^{L}\max\{|x_\ell|,\ |a_\ell|\},$
>
> where $|x|$ is the model memory, $|x_\ell|$ is the gradient memory of layer $\ell$, and $|a_\ell|$ is the activation memory. For Qwen3-0.6B, using $L=28$, hidden size $H=1024$, and $A=16$ attention heads, a conservative per-layer activation estimate is $|a_\ell| \approx \frac{4B\(8SH + AS^2)}{2^{30}}\ \text{GiB}$, where $S$ is sequence length and $B$ is batch size.  So that $M_{\mathrm{FO}}(S,B)\approx 2.22 + 28\cdot \max\left(0.0586,\ \frac{4B\(8S\cdot 1024 + 16S^2)}{2^{30}}\right)\ \text{GB}$.  Here, $2.22$ GB is the FP32 model memory of Qwen3-0.6B, and $0.0586$ GB is the FP32 gradient memory of one Transformer layer. Fixing $B=4$, this gives
>
> | Sequence Length | 256  | 512  | 1024  | 2048  |
> | --------------- | ---- | ---- | ----- | ----- |
> | Memory          | 3.86 | 5.72 | 12.72 | 37.22 |
>
> This is generally consistent with Figure 3. The  exact measured peak is higher than the simple estimate because the formula above only captures the dominant terms. In actual PyTorch training, the peak also includes temporary buffers, kernel workspaces, allocator reservation/caching, fragmentation, and additional saved intermediate tensors. Figure 3 shows that when sequence length becomes sufficiently long or batch size becomes sufficiently large, FO memory grows rapidly and eventually becomes OOM, whereas forward-only ZO methods remain feasible.
>
> **W4 (variety and size of models):**
>
> We agree that broader empirical coverage is valuable. At the same time, AGZO is not conceptually tied to a particular model scale. Its core mechanism is defined at the layer level. For trainable linear layers, AGZO exploits the structural relation between the gradient and the activation-induced subspace. This principle does not depend on the size of model.
>
> More broadly, ZO is inherently a tradeoff: it uses more optimization steps / noisier gradient estimates in exchange for a much smaller memory footprint. This makes ZO especially relevant in memory-sensitive deployment and experimentation settings, where the device can support forward inference but full FO fine-tuning is difficult or impossible due to activation memory.
>
> To directly address the reviewer’s concern on model scale, we added new experiments on Qwen3-8B. We also include OpenMathReasoning, a reasoning-oriented mathematical benchmark. These results show that the gain of AGZO persists at a larger model scale and also extends beyond lightweight classification to a reasoning-focused benchmark.
>
> |          | AGZO  | MeZO  | LOZO  |
> | -------- | ----- | ----- | ----- |
> | OpenmathReasoning | 0.640 | 0.560 | 0.580 |
> | SST2     | 0.932 | 0.916 | 0.907 |
> | COPA     | 0.820 | 0.800 | 0.790 |
> | CB       | 0.928 | 0.910 | 0.910 |

---

> > ### Author Rebuttal · Reviewer_q6JG · 2026-04-03
> >
> > Thanks for detailed comments.

---

> > > ### Author Response · Authors · 2026-04-08
> > >
> > > Thank you for the careful reading and for your positive follow-up. We are pleased that our clarifications resolved your concerns, and we will incorporate the added evidence and discussion into the final revision.

---

### Official Review · Reviewer_whtE · 2026-03-13

**Soundness:** 3
**Presentation:** 3
**Significance:** 2
**Originality:** 3
**Overall Recommendation:** 4
**Confidence:** 3

**Summary:**

This paper proposes AGZO, a zeroth-order fine-tuning method for LLMs that constructs activation-informed low-rank perturbations on the fly. The key insight is that the gradient of a linear layer is confined to the subspace spanned by its input activations, motivating perturbations restricted to this subspace rather than isotropic Gaussian directions. AGZO extracts a rank-r activation basis via lightweight power iteration during the forward pass, discards the activation matrix immediately after, and uses the basis to form low-rank perturbations. The method is theoretically shown to achieve higher expected cosine similarity to the true gradient than MeZO under activation spectral concentration. Experiments on Qwen3 (0.6B, 4B) and Pangu-1B across SuperGLUE and SQuAD benchmarks demonstrate consistent improvements over MeZO and LOZO with negligible additional memory overhead.

**Compliance With Llm Reviewing Policy:**

Affirmed.

**Final Justification:**

The rebuttal addressed most of my concerns. The distinction between AGZO and HiZOO was well explained, and the additional OPT-1.3B comparison and rank ablation results further strengthened the evaluation. However, the missing Limitations section should be addressed in the final version. Therefore, I have revised my score to 4.

**Key Questions For Authors:**

Please check the Weakness Section.

1. Figure 1(a) shows that at r=1, cosine similarity between the true gradient and its projection onto the activation subspace varies substantially across layers, with some layers as low as 0.2. Could the authors provide a rank ablation (e.g., r=1, 4, 16) to justify the uniform r=1 design choice?
2. In Figure 2, even though the similarity of AGZO increased, the order of x10^-4. Is it really sufficient to explain the performance gain through this observation?

**Limitations:**

The paper did not include a dedicated Limitations section.

**Strengths And Weaknesses:**

> Strength
1. The motivation is clear. The gradient confinement property is an interesting observation that directly motivates the design of AGZO, and Figure 1 empirically validates it.
2. The memory overhead is negligible. Storing rank-1 basis per layer is small, and Figure 3 empirically confirms that AGZO matches MeZO and LOZO in peak GPU memory across varying batch sizes and sequence lengths.
3. Overall, the motivation and methodology are well-written, and the supported theories are solid.

> Weakness
1. Missing comparison with HiZOO [1]. HiZOO is a closely related work that also leverages forward pass information to enhance ZO optimization via diagonal Hessian estimation. While AGZO and HiZOO address complementary aspects (activation-guided direction vs. curvature-aware scaling), HiZOO is neither discussed in Related Work nor included as a baseline. No comparision with HiZOO makes it difficult to assess AGZO's practical standing in the current ZO fine-tuning landscape.
2. All experiments are conducted on Qwen3 and Pangu models, which are relatively less studied in the ZO fine-tuning literature. It is unclear whether the results generalize to other architectures such as LLaMA or OPT, which are standard benchmarks in prior ZO work including MeZO [2], LOZO [3], and HiZOO.


> Reference
[1] Zhao, Yanjun, et al. "Second-order fine-tuning without pain for llms: A hessian informed zeroth-order optimizer." arXiv preprint arXiv:2402.15173 (2024).
[2] Malladi, Sadhika, et al. "Fine-tuning language models with just forward passes." Advances in Neural Information Processing Systems 36 (2023): 53038-53075.
[3] Chen, Yiming, et al. "Enhancing zeroth-order fine-tuning for language models with low-rank structures." arXiv preprint arXiv:2410.07698 (2024).

---

> ### Author Rebuttal · Authors · 2026-03-30
>
> We thank the reviewer for the valuable comments and suggestions. We are happy that the reviewer found the gradient-confinement observation and the overall methodology clear. Below we respond point by point.
>
> **W1(Discussion of HiZOO):**
>
> Our view is that HiZOO and AGZO improve ZO fine-tuning from two different and largely complementary angles. HiZOO is primarily curvature-aware. It estimates a diagonal Hessian and uses it to rescale the perturbation. In contrast, AGZO is primarily activation-aware, which changes the support of the perturbation by restricting it to an activation-informed subspace. In other words, HiZOO addresses how different coordinates should be scaled, whereas AGZO addresses which directions should be explored in the first place. Because of this distinction, we do not view AGZO as redundant with HiZOO. Rather, the two ideas are in principle compatible. To address the reviewer’s concern directly, we additionally ran HiZOO on Qwen3-0.6B in our setting:
>
> |       | AGZO  | HiZOO |
> | ----- | ----- | ----- |
> | SST2  | 0.877 | 0.854 |
> | Copa  | 0.740 | 0.680 |
> | CB    | 0.892 | 0.767 |
> | BoolQ | 0.724 | 0.692 |
>
> In these experiments, AGZO outperforms HiZOO on all four tasks. However, since HiZOO and AGZO modify different aspects of ZO optimization, a head-to-head result does not by itself settle which design principle is better. We will revise the paper to add a discussion of HiZOO as related work and clarify more clearly that AGZO should be viewed as an orthogonal refinement of forward-only ZO estimators rather than as a redundant variant of HiZOO.
>
> **W2 (generalization beyond Qwen3 / Pangu):**
>
> We selected Qwen3 and Pangu for the main experiments because, at the time of submission, they were more recent open-source LLM families. For this reason, although we did conduct early experiments on OPT during the development stage, we chose to center the main submission on newer model families.
>
> That said, we agree that showing transfer beyond these model families would further strengthen the paper. To directly address the reviewer’s concern, we provide additional OPT-1.3b results below. These results show that the gain of AGZO over prior ZO baselines is across different models.
>
> | Test Acc |  AGZO  | MEZO  | LOZO  |
> | -------- | ----- | ----- | ----- |
> | SST2     |  0.916 | 0.905 | 0.906 |
> | Copa     |  0.79  | 0.78  | 0.78  |
>
> **Q1(why choose rank-1):**
>
> Figure 1 studies a subspace property of the true gradient. It measures the cosine similarity between the true gradient and its orthogonal projection onto the activation subspace. A high value in Figure 1 means that the true gradient largely lives in that subspace. However, actual ZO optimization does not directly use the projection of true gradient on the subspace. Instead, it samples a random perturbation direction inside the space and forms a stochastic estimator. Therefore, even if the true gradient is well contained in a subspace, a single sampled perturbation within that subspace may still be only weakly aligned with the true gradient. This explains why Figure 1 can be high while Figure 2 is much lower: Figure 1 asks "does the gradient lie in this subspace?”, whereas Figure 2 asks “how well does one stochastic sampled estimator align with the gradient?”.
>
> This distinction also clarifies our rank choice. Increasing the rank enlarges the activation subspace and can capture more gradient energy, as Figure 1 suggests. But in actual ZO estimation, a larger rank also means sampling in a larger subspace, which can weaken the directional quality of a single random perturbation. Our default choice of rank 1 is therefore an efficiency-oriented operating point. It focuses exploration on the dominant activation direction while keeping both memory and estimator complexity minimal. Below we show a ablation experiments of finetuning qwen3-0.6b on sst2, which support that rank 1 is the optimal choice. We will add a more thorough ablation study in the revision.
>
> | Rank     | 1     | 4     | 16    |
> | -------- | ----- | ----- | ----- |
> | Test Acc | 0.877 | 0.870 | 0.863 |
>
> **Q2 (Interpretation of figure 2) :**
>
> In high-dimensional random-direction ZOO, very small absolute cosine values are expected even for useful estimators, because the estimator is still constructed from a single stochastic perturbation direction. Figure 2 shows that AGZO consistently improves directional fidelity over MeZO. More importantly, this relative gain is not merely geometric. It also translates into improved downstream fine-tuning performance in our end-to-end experiments. To avoid overstating the claim, we will revise the wording in the paper to emphasize higher directional fidelity relative to prior ZO baselines, rather than implying near-perfect alignment to the true gradient.

---

> > ### Author Rebuttal · Reviewer_whtE · 2026-04-03
> >
> > Thank you for the authors' effort in addressing my concerns. The rebuttal has resolved most of my questions. I have decided to increase my score to 4. Please ensure that all discussed points and limitations are integrated into the revised manuscript.

---

> > > ### Author Response · Authors · 2026-04-08
> > >
> > > Thank you for the thoughtful follow-up and for increasing your score. We are glad that our rebuttal addressed your concerns, and we will make sure that all discussed points and limitations are clearly incorporated into the revised manuscript.

---

### Decision · Program_Chairs · 2026-04-30

**Decision:**

Accept (regular)

**Comment:**

An efficient zeroth-order (ZO) finetuning method is proposed that exploits activation structure, namely that the gradient of a linear layer is confined to the subspace spanned by its input activations. Most reviewers appreciate the merit of the method (particularly regarding memory savings) and the fact that it outperforms the ZO baselines MeZO / LOZO and the experiments that were added during the rebuttal. For that reason, we propose that the paper is accepted for publication at ICML.

Reviewer GpVD raised a few concerns that should be addressed in a camera-ready version of the paper, including reporting variance across independent runs, a comparison with memory-efficient alternatives such as QLoRA or FO methods with memory-saving techniques, and an adequate discussion of absolute gradient-alignment signal.